# SUMMARIZATION PROGRAMS: INTERPRETABLE ABSTRACTIVE SUMMARIZATION WITH NEURAL MODULAR TREES

**Swarnadeep Saha, Shiyue Zhang, Peter Hase, Mohit Bansal**
Department of Computer Science
University of North Carolina at Chapel Hill
`{swarna,shiyue,peter,mbansal}@cs.unc.edu`

## ABSTRACT

Current abstractive summarization models either suffer from a lack of clear interpretability or provide incomplete rationales by only highlighting parts of the source document. To this end, we propose the *Summarization Program (SP)*, an interpretable modular framework consisting of an (ordered) list of binary trees, each encoding the step-by-step generative process of an abstractive summary sentence from the source document. A Summarization Program contains one root node per summary sentence, and a distinct tree connects each summary sentence (root node) to the document sentences (leaf nodes) from which it is derived, with the connecting nodes containing intermediate generated sentences. Edges represent different modular operations involved in summarization such as sentence fusion, compression, and paraphrasing. We first propose an efficient best-first search method over neural modules, SP-SEARCH that identifies SPs for human summaries by directly optimizing for ROUGE scores. Next, using these programs as automatic supervision, we propose seq2seq models that generate Summarization Programs, which are then executed to obtain final summaries. We demonstrate that SP-SEARCH effectively represents the generative process behind human summaries using modules that are typically faithful to their intended behavior. We also conduct a *simulation study* to show that Summarization Programs improve the interpretability of summarization models by allowing humans to better simulate model *reasoning*. Summarization Programs constitute a promising step toward interpretable and modular abstractive summarization, a complex task previously addressed primarily through blackbox end-to-end neural systems.[1]

## 1 INTRODUCTION

Progress in pre-trained language models has led to state-of-the-art abstractive summarization models capable of generating highly fluent and concise summaries (Lewis et al., 2020; Zhang et al., 2020; Raffel et al., 2020). Abstractive summarization models do not suffer from the restrictive nature of extractive summarization systems that only copy parts of the source document. However, their ability to generate non-factual content (Cao et al., 2018; Maynez et al., 2020) and their lack of clear interpretability makes it harder to debug their errors and deploy them in real-world scenarios. Towards interpretable summarization models, Jing & McKeown (1999; 2000) show that human summaries typically follow a cut-and-paste process, and propose a modular architecture involving separate operations that perform sentence extraction, sentence reduction, sentence fusion, etc. Most recent efforts on explainable abstractive summarization follow an extractive-abstractive framework that only provides supporting evidence or 'rationales' for the summary (Hsu et al., 2018; Gehrmann et al., 2018; Liu & Lapata, 2019; Zhao et al., 2020; Li et al., 2021). These models highlight words or sentences from the source document but are not able to explicitly capture the generative process of a summary, i.e., the reasoning steps performed in order to generate each summary sentence from the source document sentence(s), like sentence compression, fusion, etc. In this work, we seek to bridge

---

[1]Supporting code available at `https://github.com/swarnaHub/SummarizationPrograms`.

**Summarization Program**

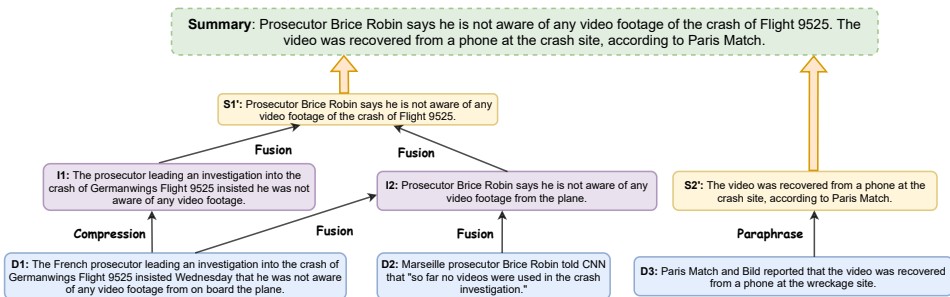

**Figure 1:** Example of a Summarization Program showing the generative process of two summary sentences (marked with labels S1' and S2' in yellow) from the three document sentences (marked with labels D1, D2, and D3 in blue) using compression, paraphrase and fusion neural modules. Edges directed from leaves to roots, and the intermediate generations are labeled with I1 and I2.

this gap by proposing a novel *Summarization Program* framework for explaining abstractive summarization, that views summarization as a systematic reasoning process over document sentences.

A Summarization Program (SP) is a modular executable program that consists of an (ordered) list of binary trees, each encoding the generative process of an abstractive summary sentence from the source document (§3). Fig. 1 shows an example. The leaves in an SP are the source document sentences (typically, only a small subset that are relevant for generating the summary). Each intermediate node represents a generation from a neural module (shown with labeled edges) which are then composed to derive the final summary sentences at the roots of the trees. We develop three neural modules for building SPs – sentence compression, paraphrasing, and fusion (Jing & McKeown, 1999; 2000) that finetune a pre-trained language model on task-specific data. We evaluate Summarization Programs by asking the following two research questions (see Fig. 2 for an overview).

1. **RQ1.** *Given a human-written abstractive summary, can we develop an algorithm for identifying a Summarization Program that effectively represents the generative process of the summary?*
2. **RQ2.** *Using the SPs identified in RQ1 as supervision, can we develop models that generate Summarization Programs as interpretable intermediate representations for generating summaries?*

We answer the first research question by automatically identifying SPs for human summaries (§4). Specifically, we develop an efficient best-first search algorithm, SP-SEARCH that iteratively applies different neural modules to a set of extracted document sentences in order to generate newer sentences, such that the ROUGE (Lin, 2004) score with respect to the gold summary is maximized by these new sentences. SP-SEARCH achieves efficiency through important design choices including maintaining a priority queue that scores, ranks, and prunes intermediate generations (appendix A). We conduct experiments on two English single document summarization datasets, CNN/DailyMail (Hermann et al., 2015) and XSum (Narayan et al., 2018) to show that SP-SEARCH outputs SPs that effectively reproduce human summaries, significantly outperforming several baselines (§6.1). Moreover, human evaluation shows that our neural modules are highly faithful,[2] performing operations they are supposed to and also generating outputs that are mostly factual to their inputs (§6.2).

We leverage SP-SEARCH to obtain oracle programs for human summaries that also serve as supervision for answering our second research question. In particular, we propose two seq2seq models for Summarization Program generation from a source document (§5, Fig. 2). In our first Extract-and-Build SP generation model, an extractive summarization model first selects a set of document sentences, which are then passed to another program-generating model. In our second Joint SP generation model, sentence extraction and SP generation happen as part of a single model. We obtain initial promising results and while state-of-the-art end2end models demonstrate better ROUGE scores, our oracle SP-SEARCH results indicate significant room for improvement in future work (§6.3).

---

[2] 'Neural module faithfulness' refers to whether the modules perform their expected operations. This is different from 'summary faithfulness' that evaluates whether a summary contains only factual claims from the source document. The latter will be referred to as 'summary factuality'.

To evaluate whether SPs improve the interpretability of summarization models, we conduct a small-scale *simulation study* (Doshi-Velez & Kim, 2017; Hase & Bansal, 2020; Zhou et al., 2022) where we ask humans to simulate model's *reasoning* by writing programs for model summaries (§6.4). We observe that after seeing model SPs, humans are able to better predict them for unseen samples, such that the executed summaries match more closely with the model summaries. Our contributions are:

- We introduce the *Summarization Program*, an interpretable modular framework for explainable abstractive summarization. Summarization Program consists of an ordered list of binary trees, each encoding the step-by-step generative process of an abstractive summary sentence through the use of different neural modules (sentence compression, paraphrasing, and fusion).
- We propose an efficient best-first search method, SP-SEARCH that identifies Summarization Programs for human-written summaries, obtaining a high ROUGE-2 of 40 on the CNN/DailyMail dataset with neural modules that are highly faithful to their intended behavior.
- We present initial Summarization Program generation models that generate SPs from a source document, which are then executed to obtain final summaries. We demonstrate that SPs improve the interpretability of summarization models by allowing humans to better *simulate* model behavior.

## 2 RELATED WORK

**Pre-trained Models for Summarization.** State-of-the-art pre-trained language models like BART (Lewis et al., 2020), T5 (Raffel et al., 2020), and PEGASUS (Zhang et al., 2020) generate summaries in an end-to-end manner. Lacking complete mechanistic understanding of transformers (Elhage et al., 2021), it remains difficult to understand the reasoning process behind the generated summary. In particular, we cannot determine whether generation follows a process similar to that of humans, e.g., using operations like selection of important information, abstraction through vocabulary generalization, sentence fusion, etc (Kintsch & van Dijk, 1978; Brown & Day, 1983; Jing & McKeown, 1999; 2000). Moreover, it is hard to determine the source of errors in summaries without knowing the reasoning path to the summary. Recent work on interpretable summarization models highlight words or sentences as rationales for the generated summaries (Hsu et al., 2018; Gehrmann et al., 2018; Liu & Lapata, 2019; Liu et al., 2019; Zhao et al., 2020; Li et al., 2021), but textual highlights do not provide a complete explanation of the summary generation process. Thus, Summarization Program is a step towards more explicit interpretability in abstractive summarization.

**Neural Module Networks.** Our work also draws inspiration from Neural Module Networks (NMN) that execute programs as learned functions composed of neural modules (Andreas et al., 2016; Hu et al., 2018; Jiang & Bansal, 2019; Gupta et al., 2020; Subramanian et al., 2020; Saha et al., 2021a; Le et al., 2022). Typically, the modules in an NMN provide attention-based explanations whose interpretability has been debated (Serrano & Smith, 2019; Wiegreffe & Pinter, 2019). Khot et al. (2021) alternatively propose Text Modular Networks that decompose multi-hop questions into sub-questions to be solved by simpler QA models. We also follow this text-in text-out paradigm for our modules but equipped to perform diverse sentence-level operations in document summarization. We evaluate our modules for 'neural module faithfulness' (Subramanian et al., 2020) to demonstrate that the SPs mostly provide a faithful interpretation of the generated summaries.

**Multi-step Reasoning over Text.** Step-by-step reasoning has received much interest as a way to explain various reasoning tasks including QA (Dalvi et al., 2021; Ribeiro et al., 2022) and natural language deduction (Bostrom et al., 2022; Saha et al., 2020; Gontier et al., 2020; Saha et al., 2021b; Tafjord et al., 2021). Summarization Programs similarly encode the reasoning steps in a summary generation process. Our SP-SEARCH method follows a forward chaining method that tries to reach a hypothesis from a set of premises (Tafjord et al., 2021; Bostrom et al., 2022), unlike backward chaining methods that do the opposite (Gontier et al., 2020; Arabshahi et al., 2021; Kalyanpur et al., 2022; Dalvi et al., 2022). In another recent line of work, chain-of-thought prompting (Nye et al., 2021; Wei et al., 2022; Wang et al., 2022) encourages LMs to generate intermediate reasoning steps before producing a final answer to a problem. However, the lack of explicit chaining between the reasoning steps and the final output may compromise the faithfulness of those steps, which have also not yet been evaluated as explanations of model behavior per se. Recent work has explored ways to force these reasoning steps to be more like deductive proofs of the final answer (Creswell & Shanahan, 2022) or instead use generations from a larger language model as silver supervision for a smaller pipeline student model (Eisenstein et al., 2022). In contrast, we aim to develop a method whose rationales (our summarization programs) exactly describe the reasoning process of the overall system

and we explicitly evaluate their faithfulness. Specifically, we generate silver programs by our search algorithm that tries to emulate the human summaries and then train models that generate programs which are further executed to obtain final summaries (and evaluated via a simulatability study).

## 3 SUMMARIZATION PROGRAM

We assume that we have a document $\mathcal{D} = \{D_i\}_{i=1}^d$ consisting of $d$ sentences and a corresponding abstractive summary $\mathcal{S} = \{S_i\}_{i=1}^s$ consisting of $s$ sentences. A Summarization Program $\mathcal{P} = \{T_i\}_{i=1}^s$ is defined as an (ordered) list of $s$ binary trees where each tree $T_i = (V_i, E_i)$ is a structured representation of the generative process of each summary sentence $S_i \in \mathcal{S}$. Fig. 1 shows an example of an SP with two trees for two summary sentences. The set of nodes $V_i$ in each tree consists of single sentences, and the edges $E_i$ are labeled with one of the neural modules $m \in \{paraphrase(\cdot), compression(\cdot), fusion(\cdot, \cdot)\}$. These modules represent operations over sentences wherein $compression(X) \rightarrow Y$ and $paraphrase(X) \rightarrow Y$ are unary operations and $fusion(X, Y) \rightarrow Z$ is a binary operation. The leaf nodes in each tree are sentences from the document $D_i \in \mathcal{D}$, and the root is a summary sentence $S_i \in \mathcal{S}$. All other nodes are intermediates sentences generated by executing a neural module (referred to as I1 and I2 in Fig. 1). An edge from a node $u \in V_i$ to a node $v \in V_i$ labeled with the module $m$ means that $v$ is generated by executing $m$ on $u$. The summary $\mathcal{S}$ is obtained by concatenating the root nodes of the trees in order. We hypothesize that the generative process of each summary sentence can be captured by composing different neural modules that operate over sentences. Following prior work on modular approaches to abstractive summarization (Jing & McKeown, 1999; 2000; Lebanoff et al., 2019; 2020b), we define the following three neural modules for building Summarization Programs.

**Fusion Module.** Sentence fusion in summarization combines information from multiple sentences (Lebanoff et al., 2020a). We finetune a BART-large model, which takes two sentences as input and outputs one fused sentence. Existing sentence fusion datasets either aim to improve discourse connections (Geva et al., 2019) or aim to fuse similar sentences from multiple documents (Brook Weiss et al., 2022). Instead, we want to fuse two disparate sentences into one sentence, which requires the model to merge related pieces and remove unimportant information. To obtain training data for such a model, we follow Lebanoff et al. (2019) to align each summary sentence from CNN/DailyMail with one to many similar and non-redundant document sentences. As our training data, we only use examples that align one summary sentence with two source sentences.

**Compression Module.** The compression module generates a compressed output of a single sentence. It involves generating a shorter sentence by preserving the essential content and the syntactic structure of the input. We finetune a BART-large model (Lewis et al., 2020) on a large parallel corpus of uncompressed and compressed sentences (Filippova & Altun, 2013).

**Paraphrase Module.** The paraphrase module generates a sentence that involves syntactic transformations or lexical paraphrasing of the input sentence. We use a publicly available PEGASUS-based (Zhang et al., 2020) paraphrase model from HuggingFace (Wolf et al., 2020).[3] In practice, we observe paraphrased outputs to frequently involve some compression as well, which we analyze in detail as part of the 'Neural Module Faithfulness' evaluation (§6.2).

## 4 RQ1: SUMMARIZATION PROGRAM SEARCH

Our first research question of interest is whether given a document $\mathcal{D}$ and a human-written summary $\mathcal{S}$, we can generate a Summarization Program $\mathcal{P}$ that best explains the generative process of the summary (see the left part of Fig. 2). We achieve this by developing an efficient best-first search method, named SP-SEARCH, outlined in Algorithm 1 in the Appendix. Conceptually, it is similar to a forward chaining algorithm in logic programming (Russell & Norvig, 2009), in which we start from a set of premises (equivalently, a small set of document sentences) and iteratively apply neural modules on them to generate newer deductions (equivalently, intermediate sentences) until the goal hypothesis (equivalently, the summary) is generated. SP-SEARCH processes each summary

---

[3]Model available at https://huggingface.co/tuner007/pegasus_paraphrase.

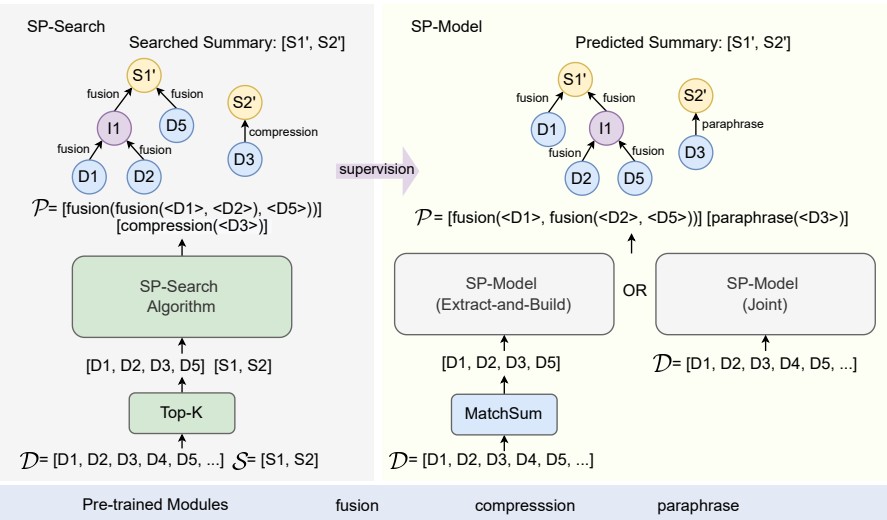

**Figure 2:** Overview of the two RQs aimed at evaluating SPs. On the left, we show our SP-SEARCH algorithm that takes a source document and a human summary as input and identifies an SP that best explains the summary (RQ1). On the right, we show our SP generation models that take a document as input and output a program, which is then executed to generate the summary (RQ2). S1 and S2 denote human summary sentences while S1' and S2' represent identified or generated summaries.

sentence separately, thereby generating a unique tree for each summary sentence. We optimize the trees for their ROUGE-L (R-L)[4] scores with respect to the gold summary sentences.

**SP-SEARCH Algorithm.** SP-SEARCH operates as follows (see Algorithm 1). For each summary sentence, it initializes a priority queue, each element $(s_1, s_2, m, h)$ of which represents a module $m$, the sentences $s_1$ and $s_2$ on which $m$ is defined ($s_2$ is empty for 'compression' and 'paraphrase' modules) and the corresponding tree height $h$. The initial elements in the queue represent possible operations only on the document sentences. Next, at each step of the search, SP-SEARCH pops an element from the queue and executes the operation over the corresponding operands to generate a new sentence. Each module generates top-5 outputs using beam search and the one with the highest R-L score is chosen. Using this new sentence, SP-SEARCH creates new elements for the priority queue by considering all potential operations involving the new sentence and other available sentences. Then, these new elements may be explored in the next steps of the search. As a best-first search method, SP-SEARCH ranks the elements in the queue with the following scoring function.

$$\text{Score}(s_1, s_2, m, h) = \max\big(\text{R-L}(s_1, S),\ \text{R-L}(s_2, S)\big) \tag{1}$$

The scoring function considers the maximum of the R-L scores with respect to generations $s_1$ and $s_2$ such that if the module $m$ is executed, it can possibly lead to a sentence that is of an even higher R-L score than its children. Intuitively, it encourages prioritizing elements in the queue that can potentially lead to higher scores. Whenever a new sentence is generated, its R-L score with the gold summary sentence is computed and accordingly, the best possible root node of the tree is updated. Upon completion of the search (when the queue becomes empty), the node with the maximum score is chosen as the root node and the tree rooted at that node is the final tree. Since performing SP-SEARCH exhaustively can be prohibitive, it is made efficient through the following design choices.

**Design Choices for Efficient SP-SEARCH.** Here we discuss the choices briefly and provide more details in Appendix A. (1) *Top-k Document Sentences.* We rank each document sentence by computing ROUGE-1 with the summary and build the SP with only the top-k document sentences as eligible source sentences. (2) *Filtering Queue Elements.* To limit the search space, SP-SEARCH defines certain heuristics for choosing the elements to be added to the queue like a compressed sentence is not compressed again, each document sentence is used at most once in a tree, etc. (3) *Priority Queue and Pruning.* SP-SEARCH maintains a fixed queue size and a ranked list of the elements according to Eq. 1 such that only the top-ranked elements are kept in the queue and the rest are pruned.

---

[4]We conducted some initial experiments to observe that optimizing for R-L leads to more consistent improvements across all ROUGE measures. See Appendix C for details.

(4) *Parent has a higher ROUGE than children.* A branch of the search is only expanded further if the new generation (parent) has a higher R-L score than the source sentence(s) (children). This greedy approach ensures that every reasoning step is a step closer to the summary. (5) *Maximum Tree Height.* SP-SEARCH chooses a maximum height of the trees. (6) *Batching Module Executions.* SP-SEARCH executes all operations in the queue together by batching at each depth of the search.

## 5 RQ2: Abstractive Summarization via Summarization Programs

Given that we have proposed SP-SEARCH for identifying SPs for human summaries, we now want to leverage the algorithm to generate training data for building SP generation models, as part of RQ2. In particular, we use SP-SEARCH to identify Summarization Programs for all training samples in the CNN/DailyMail dataset (Hermann et al., 2015). Our second research question now asks whether these identified SPs can be used as supervision for developing abstractive summarization models via the generation of SPs (see the right part of Fig. 2). Hence, we define a supervised learning problem $f : \mathcal{D} \to \mathcal{P}$ that generates a Summarization Program $\mathcal{P}$ from document $\mathcal{D}$.

**Summarization Program Generation Models.** We propose two initial models for our task. The SPs are encoded as strings such that they can be generated by a seq2seq generative model. We first label each sentence in the document with unique identifiers like '<D1>', '<D2>', etc. The Summarization Program is then represented using a nested bracketing structure composed of the modules and the sentence identifiers as shown in Fig. 2. For our first **Extract-And-Build Summarization Program Generation Model**, we hypothesize that a document can contain a large number of sentences but only a small fraction of those are typically useful for generating a summary. Given a training corpus of samples $(\mathcal{D}_k, \mathcal{P})$ where $\mathcal{D}_k$ represents the set of $k$ document sentences on which the SP $\mathcal{P}$ is built, we finetune BART such that it can generate a Summarization Program given a small number of document sentences. During inference, given a document, we first extract a set of relevant sentences using a state-of-the-art extractive summarization model, MatchSum (Zhong et al., 2020), which are then used to generate the program. While this model simplifies the learning problem by separating out the sentence extraction phase from the program building phase, relevant sentence extraction is an essential first step towards generating good SPs. Our next **Joint Summarization Program Generation Model** aims to solve sentence extraction and SP generation as part of a single model. We finetune BART on the parallel corpus of document and Summarization Programs and the model learns to directly generate the program given an entire document.

**Summary Generation from an SP.** We parse and execute a generated SP through iterative inference over the neural modules to obtain the final summary. We check for well-formedness of the SP by ensuring that the generated sequence (1) has balanced parentheses, (2) does not contain any out-of-vocabulary token (besides module names, sentence identifiers, and parentheses), and (3) has consistent number of operands for an operation. During inference, we decode up to top-k SPs using beam search and execute the first well-formed one. When none of the top-k outputs is well-formed (<1% of the samples), we generate the corresponding extractive summary.

## 6 Experiments

We experiment with two English single-document summarization datasets, CNN/DailyMail (Hermann et al., 2015) and XSum (Narayan et al., 2018). We discuss CNN/DM results below and point readers to Appendix F for XSum results.

### 6.1 RQ1 Results: Can SP-Search Represent the Summarization Process?

Our first set of experiments is aimed at answering our first RQ (as discussed in §4) where the goal is to analyze how well the Summarization Programs identified by SP-SEARCH represent the human-written summaries. We consider two variants of SP-SEARCH – (1) SP-SEARCH Top-1 in which each module generates only one output, and (2) SP-SEARCH where each module generates Top-5 outputs (via beam search) and one with the best R-L is chosen.

**Evaluation Metrics.** We use ROUGE to measure similarity between the SP-SEARCH summaries and the human summaries. We also compute factuality of the SP-SEARCH summaries with respect of the gold summaries using a SOTA factuality metric, QuestEval (Scialom et al., 2021). Note that

while factuality is typically measured against the source document, RQ1 requires us to evaluate how factual the SP-SEARCH summary is to the reference summary (that the SPs are aiming to reproduce).

**Experimental Design.** We conduct experiments with 1000 random validation samples.[5] We set the number of initial document sentences ($k$) to 4, the maximum queue size ($Q$) to 20, maximum height of the trees ($H$) to 2 and the decoding strategy for each module to beam search with beam size of 5. Our choice of hypeparameters is based on a comprehensive study of the trade-off between ROUGE scores and average search time, as presented in Appendix C. As baselines, we consider the following extractive and abstractive oracles. (1) **Random SP.** For each summary sentence in the gold summary, we randomly sample a tree structure (from those obtained through SP-SEARCH) and execute it with randomly chosen leaf nodes from the set of top-k sentences. (2) **Top-4 Sentences.** Our next baseline is an extractive summarization system with the top-4 document sentences ranked according to ROUGE-1 scores with the gold summary. (3) **BART-Oracle.** We also compare with two BART-oracle models. In one, we generate top-10 summaries using beam search. In another, we sample 10 summaries with multinomial sampling and the summary with the highest ROUGE-L score with the gold summary is chosen. (4) **SP-SEARCH Leaves.** Our final baseline is an SP-SEARCH variant where we only consider the leaf nodes (document sentences). This represents an extractive summary with *upto*[6] top-4 sentences. Since the trees are built on top of these nodes, a better ROUGE score for SP-SEARCH will indicate that these leaf nodes or document sentences are being composed in a way such that the resultant summary is more similar to the human summary. (5) **Ablations of Modules.** These are SP-SEARCH variants where each module is removed.

**Results.** As shown in Table 1, SP-SEARCH generates summaries that better emulate gold summaries by obtaining an R-2 of 40, a significant[7] 20 points improvement ($p < 0.001$) over the SP-SEARCH leaves. The random SP baseline uses the same pre-trained modules as SP-SEARCH but only obtains an R-2 of 15, demonstrating that arbitrary composition of these modules is much less effective than the searched programs. The BART-oracle models exhibit less diversity between the generations, as reflected by their lower ROUGE scores. This demonstrates the utility of a tree structure in an SP. Finally, we observe that each of our neural modules plays a significant

| | R1 / R2 / RL / RLsum | QEval |
|---|---|---|
| Random SP | 37.54 / 15.18 / 25.65 / 33.42 | 40.50 |
| Top-4 Sentences | 35.55 / 17.77 / 25.02 / 32.42 | 50.56 |
| BART-Oracle (Beam) | 40.07 / 19.57 / 32.04 / 37.06 | 43.25 |
| BART-Oracle (Sample) | 42.07 / 19.44 / 33.56 / 39.04 | 45.09 |
| SP-SEARCH Leaves | 39.15 / 20.01 / 27.94 / 35.58 | 50.53 |
| SP-SEARCH Top-1 | 56.92 / 34.34 / 52.15 / 54.80 | 52.32 |
| SP-SEARCH | **61.88 / 40.11 / 58.46 / 60.32** | **54.48** |
| - Paraphrase | 60.08 / 38.28 / 56.36 / 58.29 | 53.80 |
| - Compression | 59.50 / 37.44 / 55.45 / 57.66 | 53.85 |
| - Fusion | 53.73 / 31.30 / 48.90 / 51.73 | 48.78 |

**Table 1:** RQ1 – ROUGE scores for SP-SEARCH summaries and oracle extractive, abstractive baselines. Final column compares the factuality of the summaries with respect to the *gold summaries* through a factuality metric, QuestEval.

role in constructing more accurate SPs. Removing the fusion module leads to the highest drop in ROUGE which suggests that fusing information from multiple sentences is one of the core operations in human summaries. SP-SEARCH also obtains the best QuestEval score, even though it does not specifically optimize for it, suggesting that the summaries are not only similar to the human summaries (as measured by ROUGE) but also have high factual overlap with the human summaries. In summary, SP is an effective way of representing the generative process of abstractive summaries, outperforming random SPs, extractive oracles and unstructured abstractive oracles.

## 6.2 RQ1 RESULTS: NEURAL MODULE FAITHFULNESS EVALUATION

Higher ROUGE and factuality scores obtained by SP-SEARCH with respect to the reference summaries do not guarantee that SPs provide a faithful interpretation of the generated summaries. §6.1 specifically evaluates the final summaries but not the intermediate steps used to generate them. In fact, prior work on Neural Module Networks has shown that faithfulness may not always be guaranteed for neural modules and that incorrect intermediate outputs can still lead to correct final outputs (Subramanian et al., 2020).

---

[5]Due to the expensive nature of SP-SEARCH, we experiment with a random subset of the validation set.

[6]SP-SEARCH can ignore certain top-4 sentences if including them do not lead to higher ROUGE scores.

[7]We use a non-parametric bootstrap test (Efron & Tibshirani, 1994) for significance testing.

Hence, we also evaluate the faithfulness of the three modules by conducting a human study (with 2 experts having knowledge of NLP) for 20 SPs. These constitute a total of 107 samples, including 54 fusion, 25 paraphrase, and 28 compression operations. Table 2 shows the results (with more details in Appendix E). The diagonal entries in the table demonstrate how often a module performs its intended behavior and the high values of 0.8-0.98 suggest

|       | Comp | Para | Fusion | Non-Factual |
|-------|------|------|--------|-------------|
| Comp  | 0.98 | 0.05 | -      | 0.00        |
| Para  | 0.68 | 0.90 | -      | 0.04        |
| Fusion| 0.86 | 0.45 | 0.80   | 0.20        |

**Table 2:** RQ1 – Neural Module Faithfulness evaluation of SPs. Entry (i,j) shows how often a module 'i' in a row performs the operation 'j' in a column. The final column shows the fraction of non-factual outputs.

that our modules are highly faithful. Interestingly, the values in the last column demonstrate that compression and paraphrase modules almost never generate non-factual outputs while our fusion module is more prone to that (about 20%), indicating room for improvement in the fusion module.

### 6.3 RQ2 Results: Evaluation of Summarization Program Generation Models

Our next set of experiments addresses our second RQ (as discussed in §5) on evaluating Summarization Program Generation models in terms of their summary generation capabilities.

**Experimental Design.** We compare our SP generation models (Joint and Extract-and-Build) with the following baselines and oracles on the CNN/DailyMail test set. As baselines, we consider (1) a **SOTA Extractive Summarization Model, MatchSum (Zhong et al., 2020)** that we also use to extract initial document sentences for our Extract-and-Build SP generation model, (2) **SOTA Abstractive Summarization Models, BART (Lewis et al., 2020) and PEGASUS (Zhang et al., 2020)**, and (3) **Random SP Models** that randomly sample the number of summary sentences from {1,2,3,4}, and for each summary sentence, randomly sample and execute a tree structure with randomly chosen leaf nodes from all or top-k document sentences. As oracles, we consider the same models introduced in §6.1 for RQ1. They provide an estimate of the upper bound of SP models.

**Evaluation Metrics.** Besides ROUGE, we also evaluate the factuality of the generated summaries (with respect to the source document) using QuestEval that is shown to correlate well with humans.

**Results.** Table 3 shows the results. We observe that our models obtain R-2 and R-L scores of 16 and 26 respectively. The Extract-and-Build model performs slightly better than our Joint model, possibly due to the former generating programs over a good initial set of document sentences. Both of our models outperform the Random SP baselines, demonstrating that they learn useful patterns of which neural modules should act on which document sentences. Compared to SOTA abstractive models,

|                                    | R1 / R2 / RL / RLsum          | QEval |
|------------------------------------|-------------------------------|-------|
| MatchSum (Zhong et al., 2020)      | 43.98 / 20.66 / 28.72 / 40.02 | 63.22 |
| BART (Lewis et al., 2020)          | 44.19 / 21.29 / 31.06 / 41.04 | 59.15 |
| PEGASUS (Zhang et al., 2020)       | 44.16 / 21.55 / 31.38 / 41.01 | 58.83 |
| Random SP (Joint)                  | 24.47 / 6.48 / 15.60 / 22.07  | 40.73 |
| Random SP (Extract-and-Build)      | 31.61 / 12.69 / 21.84 / 28.88 | 52.83 |
| SP Model (Joint)                   | 37.66 / 16.00 / 25.68 / 34.45 | 56.88 |
| SP Model (Extract-and-Build)       | 37.23 / 16.40 / 26.48 / 34.18 | 55.95 |
| Top-4 Sentences (Oracle)           | 45.35 / 24.60 / 31.30 / 41.92 | 60.00 |
| SP-Search Leaves (Oracle)          | 47.28 / 25.92 / 32.73 / 43.70 | 59.60 |
| SP-Search Top-1 (Oracle)           | 58.17 / 35.31 / 51.87 / 55.96 | 54.72 |
| SP-Search (Oracle)                 | 62.69 / 40.58 / 58.02 / 61.10 | 54.11 |

**Table 3:** RQ2 – Comparison of our SP generation models with state-of-the-art extractive and abstractive summarization models, random SP baselines and oracle systems with our SP-Search variants on the CNN/DM test set. All methods are compared based on ROUGE and a state-of-the-art factuality metric, QuestEval (Scialom et al., 2021).

our models' interpretability (as discussed in §6.4) comes at the cost of some drop in performance (about 5 points in R-2). However, the oracle results suggest that our models provide good starting points for better SP generation models and that there is substantial room for improvement in future work (e.g., the SP-Search Top-1 model obtains an R-2 of 35, leaving a large gap of 19 R-2 points to oracle SPs). We hope the community will explore and improve upon our SP generation models, as a way to make progress on the important and challenging task of developing interpretable abstractive summarization models. We observe a similar trend for the QuestEval scores where our

models largely outperform the random SP baselines, while showing worse performance than BART and PEGASUS, possibly because of the cascading effect of compositional generation via SPs.

**Discussion.** BART and PEGASUS optimize for gold summaries while our SP models optimize for SP-Search summaries. The SP models, by themselves, are simple and effective – when the generated summaries are evaluated using *SP-Search summaries* as targets, we obtain a comparable R-2 of 21 to SOTA models. This shows that retraining our models on even better oracle programs by incorporating more modules to SP-SEARCH or further enhancing the existing ones can help close the gap to SOTA models. We hope the community will explore and improve upon our methods, as a way to make progress on the challenging task of developing interpretable summarization models.

## 6.4 RQ2 RESULTS: INTERPRETABILITY EVALUATION OF SP VIA SIMULATION STUDY

We ask if SPs improve the interpretability of a summarization model. In particular, we are interested in evaluating model *simulatability*, a measure of whether a person can predict model behavior on new inputs (Doshi-Velez & Kim, 2017). Similar to Zhou et al. (2022), we are specifically interested in model *reasoning*, as represented by the programs that our models generate and execute. The primary motivation for predicting model reasoning is that it is what we want to better understand by virtue of model explanations, and simulation studies that focus on predicting final outputs do so only to show that users have a good mental model of model reasoning (Hase & Bansal, 2020).

**Study Design.** With 3 experts (authors) having prior knowledge in NLP and explainability, we design and carry out a small-scale human study that tests whether presenting humans with Summarization Programs improves their ability to predict SPs on future unseen data. Human subjects are first provided with some *test* documents and corresponding model summaries (from our Explain-and-Predict SP model) and are tasked with predicting the model's Summarization Programs. Next, in a *training* phase, they are shown

|  | R1 / R2 / RL / RLsum |
|---|---|
| Before Expl. | 74.37 / 62.79 / 71.62 / 72.54 |
| After Expl. | **78.03 / 66.80 / 72.37 / 74.69** |

**Table 4:** RQ2 – Comparison of mean ROUGE scores between our model summaries and their human simulations before and after being trained with SPs.

model-generated SPs for a few *training* samples and are then asked again to perform the same task with the same set of *test* examples. The study is conducted with 10 *training* examples and 31 *test* examples. We say that an SP is more representative of the model summary if executing it with our pre-trained modules generates a summary that is closer to the model summary. This is a measure of program similarity since variance in the SPs may make graph similarity measures inappropriate (e.g., compression followed by paraphrase and vice versa may generate similar outputs).

**Results.** The results are presented in Table 4. In general, high ROUGE scores suggest that, when models are forced to generate summaries via SPs, their reasoning becomes quite predictable a priori. We also see about 4 points improvement in ROUGE-2, with 61% of samples showing an improved R-2 score (statistically significant at $p < 0.001$), 10% of samples with a drop in R-2, and remaining being ties. This suggests that SPs are potentially good explanations of model reasoning, such that humans can generalize across model reasoning patterns after being given the explanations.

## 7 DISCUSSION AND CONCLUSION

We proposed the Summarization Program, a novel framework for interpretable abstractive summarization. We demonstrated its effectiveness by developing SP-SEARCH that identifies summarization programs for human summaries with highly faithful neural modules and SP models that produce summaries from source documents. Two most common forms of errors in the generated SPs include (1) redundant or longer paths in an SP, and (2) fusion module generating non-factual sentences or ignoring one of its source sentences. Two other notable issues, arising out of the independence assumption of summary sentences, are (1) final summary sentences having overlapping information, and (2) incoherence between consecutive sentences. One way to improve this is to add a 'coherence' module on top of the root nodes before generating the final summary. We build SPs using sentences as the fundamental content unit (nodes) due to the relative ease of defining and training neural modules on sentences and the availability of large-scale training data. Summarization may also involve other text manipulation operations that are not fully captured by our modules but our framework allows easy inclusion of other modules.

## ETHICS STATEMENT

Despite the recent success of pre-trained language models in abstractive summarization, their lack of explainability still remains a major concern and we hope that Summarization Programs prove to be an important step in bridging that gap. That said, summarization is inherently a subjective task and existing summarization datasets also significantly vary in terms of their stylistic features like abstractiveness, length, and specificity (Goyal et al., 2022). Hence, more future work is needed to understand the general applicability of our neural modules and how effective they are in encoding different kinds of summaries. Broadly put, Summarization Program is a case study for breaking a complex NLP problem down into sub-problems and then solving them through neural modules without having access to intermediate supervision. We fine-tune language models for building modules and language models can be prone to generating unwanted content (Weidinger et al., 2021). However, since each module is focused on one particular skill, that should help limit the negative impact and provide users with more control, compared to end-to-end models. Summarization Programs are also specifically designed to trace the origin of any toxic or hallucinated content in the generated summaries.

## REPRODUCIBILITY STATEMENT

To encourage reproducibility, we make our source code publicly available. The details of our SP-SEARCH algorithm are shown in Algorithm 1. The hyperparameters for RQ1 and RQ2 are discussed in Appendix C and Appendix D respectively. The CNN/DailyMail and XSum datasets are also publicly available at https://huggingface.co/datasets/cnn_dailymail and https://huggingface.co/datasets/xsum respectively.

## ACKNOWLEDGEMENTS

We thank the reviewers for their valuable feedback. We also thank Archiki Prasad and Prateek Yadav for their help with neural module faithfulness annotations, and David Wan for help with factuality evaluation. This work was supported by NSF-CAREER Award 1846185, NSF-AI Engage Institute DRL-2112635, Google Ph.D. Fellowship, and Bloomberg Data Science Ph.D. Fellowship.

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

| Tree Structure | $H$ | Freq (in %) |
|---|---|---|
| compression ($\cdot$) | 1 | 8 |
| compression ( fusion ($\cdot$, $\cdot$) ) | 2 | 8 |
| fusion ( fusion ($\cdot$, $\cdot$) fusion ($\cdot$, $\cdot$) ) | 2 | 7 |
| ($\cdot$) | 0 | 7 |
| fusion ( compression ($\cdot$) fusion ($\cdot$, $\cdot$) ) | 2 | 6 |
| paraphrase ( compression ($\cdot$) ) | 2 | 6 |
| paraphrase ( fusion ($\cdot$, $\cdot$) ) | 2 | 6 |
| fusion ( fusion ($\cdot$, $\cdot$) compression ($\cdot$) ) | 2 | 5 |
| fusion ( fusion ($\cdot$, $\cdot$) ) | 2 | 5 |
| fusion ( compression ($\cdot$) ) | 2 | 5 |

**Table 5:** Top 10 tree structures for the human-written summaries in the training corpus of CNN/DM. For clarity, each tree structure is accompanied with the corresponding tree height and the frequency (in percentage). An empty structure of "($\cdot$)" represents a singleton node.

## A    DESIGN CHOICES FOR EFFICIENT SP-SEARCH (CONTINUED FROM §4)

SP-SEARCH is outlined in Algorithm 1. It is made efficient through important design choices as discussed below. For clarity, we do not show all of these choices in the Algorithm outline.

**Top-k Document Sentences.** The search space grows exponentially at each depth (because of the fusion operation), but one way to limit the growth is to ensure that SP-SEARCH starts with a small number of document sentences. A summary is typically constructed by using information from only a small number of sentences from the document. Hence, we rank each document sentence by computing the ROUGE-1 score with the summary and build the Summarization Program with only the top-k document sentences ('k' being a small number) as eligible source sentences.

**Filtering Queue Elements.** The search space is also dependent on how many elements are added to the queue at each step and how many of those are expanded further. Note that whenever a new sentence is generated via a module, it can potentially fuse with all previous generations to create new queue elements. Since doing this exhaustively increases the search space, SP-SEARCH defines certain heuristics for choosing the elements that will be added to the queue: (1) a sentence that has been compressed once is not compressed again, (2) a sentence that has been paraphrased once is not paraphrased again, (3) each document sentence is used at most once in a tree, (4) two sentences are not fused if they are intermediate generations from the same sentence, and (5) since fusion is not a symmetric operation and can lead to different generations based on the order, sentence fusion happens keeping the temporal order of the sentences in the document intact.

**Priority Queue and Pruning.** SP-SEARCH performs an additional step of pruning for the elements that are added to the queue. It maintains a fixed queue size and while expanding the elements in the queue in a best-first manner, it maintains a ranked list of the elements according to Eq. 1 such that only the top-ranked elements are kept in the queue and the rest are pruned.

**Parent has a higher ROUGE than children.** Each module generates multiple outputs through beam search and SP-SEARCH ensures that the best output has a higher R-L score than the nodes on which the module is defined. If this is not the case, the corresponding branch of the search is not expanded further. This constraint generalizes to the property that every node in a Summarization Program will have a higher R-L score (with respect to the summary sentence) than all other nodes in the subtree rooted at that node. Conceptually, this greedy approach ensures that every reasoning step in a Summarization Program is a step closer to the summary (according to a scoring function).

**Maximum Tree Height.** SP-SEARCH chooses a maximum height of the trees, beyond which the nodes are not expanded further during the search.

**Batching Module Executions.** Instead of executing each module separately, which requires a forward pass over a neural model and can be time-consuming, SP-SEARCH executes all operations in the queue together by batching at each depth of the search.

---

**Algorithm 1:** SP-SEARCH Algorithm

---

**Input:** Top-k Document Sentences $\mathcal{D}_k$, Summary $\mathcal{S}$, Modules $\{\mathcal{M}_c, \mathcal{M}_p, \mathcal{M}_f\}$, Maximum Height $\mathcal{H}$,
  Maximum Queue Size $\mathcal{Q}$, Number of Generations $\mathcal{G}$, Scoring Function $Score$
**Output:** Summarization Program $\mathcal{P}$

1 **function** SP-SEARCH($\mathcal{D}_k, \mathcal{S}, \{\mathcal{M}_c, \mathcal{M}_p, \mathcal{M}_f\}, \mathcal{H}, \mathcal{Q}, \mathcal{G}, Score$)
2  $\quad \mathcal{P} = []$ ▷ Initialize Summarization Program.
3  $\quad$ **forall** $S \in \mathcal{S}$ **do**
4  $\quad\quad queue \leftarrow [], height \leftarrow 1$ ▷ Initialize queue of items and tree height.
5  $\quad\quad maxRouge \leftarrow 0, bestRoot \leftarrow \phi$ ▷ Initialize a tree with maximum ROUGE.
6  $\quad\quad$ **forall** $\mathcal{D}_1 \in \mathcal{D}_k$ **do**
7  $\quad\quad\quad currRouge \leftarrow \text{ComputeRouge}(\mathcal{D}_1, S)$ ▷ Compute ROUGE with leaf nodes.
8  $\quad\quad\quad$ **if** $currRouge > maxRouge$ **then**
9  $\quad\quad\quad\quad maxRouge \leftarrow currRouge, bestRoot \leftarrow \text{Node}(\mathcal{D}_1)$ ▷ Update ROUGE and root.
10 $\quad\quad\quad queue \leftarrow queue \cup \{(\mathcal{D}_1, \phi, \mathcal{M}_c, height)\}$ ▷ Add compression items.
11 $\quad\quad\quad queue \leftarrow queue \cup \{(\mathcal{D}_1, \phi, \mathcal{M}_p, height)\}$ ▷ Add paraphrase items.
12 $\quad\quad\quad$ **for** $\mathcal{D}_2 \in \mathcal{D}_k$ **do**
13 $\quad\quad\quad\quad queue \leftarrow queue \cup \{(\mathcal{D}_1, \mathcal{D}_2, \mathcal{M}_f, height)\}$ ▷ Add fusion items.

14 $\quad\quad prevHeight \leftarrow 0, visited \leftarrow []$
15 $\quad\quad$ **while** $|queue| > 0$ **do**
16 $\quad\quad\quad height \leftarrow queue[0].height$ ▷ Get height of the top queue item.
17 $\quad\quad\quad$ **if** $height > \mathcal{H}$ **then**
18 $\quad\quad\quad\quad$ continue ▷ Stop expanding beyond max height.
19 $\quad\quad\quad$ **if** $height \neq prevHeight$ **then**
20 $\quad\quad\quad\quad prevHeight \leftarrow height$
21 $\quad\quad\quad\quad queue \leftarrow \arg\max_{q \in queue}(queue, \mathcal{Q}, Score)$ ▷ Rank and prune the queue.
22 $\quad\quad\quad\quad gens \leftarrow \text{Execute}(queue, \{\mathcal{M}_c, \mathcal{M}_p, \mathcal{M}_f\}, \mathcal{G})$ ▷ Execute all operations in
    queue.
23 $\quad\quad\quad newNode \leftarrow queue.\text{pop}()$ ▷ Pop the top item from the queue.
24 $\quad\quad\quad newGen \leftarrow gens[newNode]$ ▷ Get the newly generated sentence.
25 $\quad\quad\quad newRouge \leftarrow \text{ComputeRouge}(newGen, S)$ ▷ Get rouge with new sentence.
26 $\quad\quad\quad$ **if** $newRouge > maxRouge$ **then**
27 $\quad\quad\quad\quad maxRouge \leftarrow newRouge, bestRoot \leftarrow \text{Node}(newGen)$ ▷ Update root.
28 $\quad\quad\quad$ **if** $newGen \notin visited$ **then**
29 $\quad\quad\quad\quad visited \leftarrow visited \cup \{newGen\}, height \leftarrow height + 1$
30 $\quad\quad\quad\quad queue \leftarrow queue \cup \{(newGen, \phi, \mathcal{M}_c, height)\}$ ▷ Add compression items.
31 $\quad\quad\quad\quad queue \leftarrow queue \cup \{(newGen, \phi, \mathcal{M}_p, height)\}$ ▷ Add paraphrase items.
32 $\quad\quad\quad\quad$ **forall** $prevGen \in visited$ **do**
33 $\quad\quad\quad\quad\quad queue \leftarrow queue \cup \{(newGen, prevGen, \mathcal{M}_f, height)\}$ ▷ Add fusion
      items.

34 $\quad\quad T_S = \text{ConstructTree}(bestRoot)$ ▷ Get Tree by backtracking from best root.
35 $\quad\quad \mathcal{P} \leftarrow \mathcal{P} \cup \{T_S\}$ ▷ Append new tree to the Summarization Program.
36 $\quad$ **return** $\mathcal{P}$

---

# B  ANALYSIS OF SP-SEARCH SUMMARIZATION PROGRAMS FOR CNN/DM

We observe that 68% of the summaries in CNN/DM have 3 or 4 summary sentences and, equivalently, the corresponding Summarization Programs have 3 or 4 trees. Note that while we initialize SP-SEARCH with the Top-4 document sentences, a Summarization Program may choose to ignore some of these sentences if including them does not lead to higher ROUGE scores. Upon analysis, we find that 73% of the programs are constructed using all four initial sentences, 23% are constructed with three sentences and 3% have two sentences. We also note that the trees can have any height up to the maximum defined height of 2. A tree of height 0 represents a singleton node with a single document sentence. Thus, a Summarization Program with only singleton nodes represents an extractive summary. Overall, we observe that our trees have as many as 20 different structures. Table 5 shows the top 10 tree structures in our corpus. As an example, a tree structure of "compression (

| $k$ | $Q$ | $D$ | $H$ | ROUGE (R1/R2/RL) | $T$ |
|---|---|---|---|---|---|
| 3 | 20 | beam (5) | 2 | 59.36 / 37.38 / 55.59 | 21.09 |
| 4 | 20 | beam (5) | 2 | 61.88 / 40.11 / 58.46 | 29.58 |
| 5 | 20 | beam (5) | 2 | 63.72 / 41.99 / 60.46 | 35.80 |
| 4 | 10 | beam (5) | 2 | 61.21 / 39.06 / 57.46 | 15.38 |
| 4 | 20 | beam (5) | 2 | 61.88 / 40.11 / 58.46 | 29.58 |
| 4 | 30 | beam (5) | 2 | 62.32 / 40.57 / 58.97 | 41.61 |
| 4 | 20 | beam (5) | 1 | 56.45 / 34.16 / 51.34 | 13.58 |
| 4 | 20 | beam (5) | 2 | 61.88 / 40.11 / 58.46 | 29.58 |
| 4 | 20 | beam (5) | 3 | 65.00 / 43.81 / 62.36 | 34.68 |
| 4 | 20 | greedy | 2 | 52.87 / 30.12 / 47.08 | 26.85 |
| 4 | 20 | beam (1) | 2 | 56.91 / 34.34 / 52.15 | 28.56 |
| 4 | 20 | beam (5) | 2 | 61.88 / 40.11 / 58.46 | 29.58 |

**Table 6:** Analysis of ROUGE scores (R1/R2/RL) and average search time (in seconds) for SP-SEARCH on 1000 validation samples in CNN/DM with different search hyperparameters. k = Number of extracted document sentences. Q = Maximum queue size. D = Decoding Strategy from the modules. H = Maximum height of the tree. T = Average search time in seconds. Search times are computed on a single RTX 2080 Ti GPU. Beam(5) refers to beam search with beam size 5 and best of top-5 generations from each module.

| Opt. Metric | R1 | R2 | RL | RLsum |
|---|---|---|---|---|
| R1 | **64.28** | 38.40 | 54.36 | 57.68 |
| R2 | 59.11 | **41.85** | 53.28 | 55.97 |
| RL | 61.88 | 40.11 | **58.46** | **60.32** |

**Table 7:** Comparison of ROUGE scores for the summaries generated by SP-SEARCH based on the optimization metric. Optimizing for R-L leads to more consistent results across all ROUGE metrics.

fusion $(\cdot, \cdot)$ )" represents a tree of height 2 in which two document sentences are first fused and then the resultant sentence is compressed.

## C    SP-SEARCH HYPERPARAMETERS

In order to analyze the effect of different hyperparameters in SP-SEARCH, we compare the trade-off between ROUGE[8] scores and average search time per sample (on a single RTX 2080 Ti GPU) by varying the number of initial document sentences ($k$), the queue size ($Q$), decoding strategy ($D$) and the maximum height of the trees ($H$) in Table 6. We observe that increasing the number of document sentences $k$ beyond 4 leads to some improvements in ROUGE scores. However, this comes at the cost of increased search time. Increasing the queue size ($Q$) also leads to minor improvements but again suffers from increased search time. While increasing the maximum tree height ($H$) to 3 results in better ROUGE scores, on inspection, we observe that it happens primarily due to low-quality fusion of two arbitrary sentences that may not always lead to better programs. Finally, beam search performs better than greedy decoding, and leveraging the best of the top-5 generations from each module improves the results further with almost no increase in search time. Overall, at a moderate search time of less than 30 seconds/sample on a single RTX 2080 Ti GPU, SP-SEARCH obtains a 20 point increase in R-2 compared to our extractive baseline with leaf nodes (document sentences).

SP-SEARCH builds Summarization Programs by optimizing for ROUGE-L because of its better overall performance across all ROUGE metrics. As shown in Table 7, optimizing instead for R-1 and R-2 leads to slightly better R-1 and R-2 respectively but overall R-L performs better. In general,

---

[8]We use the HuggingFace implementation of ROUGE at https://github.com/huggingface/datasets/blob/main/metrics/rouge/rouge.py.

| IS Metric | R1 | R2 | RL | RLsum |
|-----------|-------|-------|-------|-------|
| R1 | 61.88 | 40.11 | 58.46 | 60.32 |
| R2 | **62.07** | **41.61** | **59.05** | **60.76** |
| RL | 61.79 | 40.36 | 58.66 | 60.38 |

**Table 8:** Comparison of ROUGE scores for the summaries generated by SP-SEARCH based on the metric used for choosing the initial sentences (IS). The optimization metric is set to R-L for all.

|  | Compression | Paraphrase | Fusion | Non-Factual |
|------------|-------------|------------|--------|-------------|
| Compression | 0.98±0.01 | 0.05±0.02 | - | 0.00±0.00 |
| Paraphrase | 0.68±0.04 | 0.90±0.05 | - | 0.04±0.00 |
| Fusion | 0.86±0.01 | 0.45±0.03 | 0.80±0.03 | 0.20±0.05 |

**Table 9:** RQ1 – Neural Module Faithfulness evaluation of Summarization Programs. Each entry (i,j) shows the fraction of time a module 'i' in a row performs the operation 'j' in a column. The final column shows how often a module generates non-factual outputs. We report the mean and variance of the scores between the two annotators.

we find that the different ROUGE metrics may not correlate well with each other when optimizing for one of these.

SP-SEARCH uses ROUGE-1 scores to select the candidate sentences for building the Summarization Programs. We also experiment with other ROUGE metrics in Table 8 and observe that our algorithm is fairly robust to the choice of metric. R-2 obtains slightly better results than R-1 and R-L but overall, all metrics still obtain sufficiently high oracle results of 40-41 points in R-2 through SP-Search, independent of the selection metric.

## D  SP MODEL HYPERPARAMETERS

We build our models on top of the HuggingFace transformers library (Wolf et al., 2020). All models are trained for $40000$ steps with a batch size of 16, learning rate of $3 * 10^{-5}$ and warmup steps of $500$. We set the maximum input length to $512$ and maximum generation length to $100$. During inference, we generate up to Top-10 Summarization Programs with beam search and output the first well-formed program. We also set the minimum generation length to $10$ to prevent the model from generating too short sequences and repetition penalty to $2$. Program execution from the SPs is performed with the same set of hyperparamaters for each module as used during SP-SEARCH.

## E  RQ1 RESULTS: NEURAL MODULE FAITHFULNESS EVALUATION ON CNN/DM (CONTINUED FROM §6.1)

We discuss the faithfulness evaluation of our modules below.

**Study Design.**  We would attribute each sentence to exactly one of the modules, but since our modules are generative, we do not know if each module performs only and exactly its named function. For example, fusing two sentences into a fluent sentence may in practice also involve some paraphrasing or compression. Hence, we evaluate module faithfulness by analyzing how often a module $m_i$ performs the role of a module $m_j$. Two expert annotators, with knowledge of NLP, annotate each intermediate generation (from a module) and assign a binary label against each of the modules that could lead to that output. Additionally, the annotators also label each generation for whether it contains non-factual content (that cannot be verified from the sentence(s) used to generate it). The study is conducted with 20 Summarization Programs, consisting of a total of 107 samples, with 54 fusion, 25 paraphrase, and 28 compression operations.

**Results.**  Table 9 shows the results. The diagonal entries in the table demonstrate how often a module performs its intended behavior and the high values of 0.8-0.98 suggest that our modules

|  | R1 | R2 | RL | RLsum |
|---|---|---|---|---|
| Lead 1 | 15.88 | 1.54 | 11.71 | 11.81 |
| SP-SEARCH Leaves | 25.10 | 5.89 | 16.48 | 19.44 |
| Top-1 Sentence | 29.77 | 7.21 | 20.85 | 21.13 |
| Random SP | 28.96 | 9.04 | 22.38 | 22.41 |
| BART-Oracle (Sample) | 49.23 | 25.82 | 42.31 | 42.30 |
| SP-SEARCH Top-1 | 47.72 | 24.83 | 43.50 | 43.50 |
| SP-SEARCH | **51.86** | **29.60** | **48.48** | **48.51** |

**Table 10:** RQ1 – Comparison of ROUGE scores for the SP-SEARCH summaries with different oracle extractive and abstractive baselines on the XSum dataset.

are highly faithful. In about 20% of the cases, a fusion operation may not involve any fusion. We also observe that some form of compression frequently happens as part of the paraphrasing and fusion modules, as shown in the first column of the table. Similarly, the fusion outputs also tend to involve some compression and paraphrasing operations. Interestingly, the values in the last column of the table demonstrate that compression and paraphrase modules almost never generate non-factual outputs while our fusion module is more prone to that (about 20%), indicating room for improvement in the fusion module.

## F  RESULTS ON XSUM DATASET (CONTINUED FROM §6)

To further test the generalizability of Summarization Programs, we conduct experiments on another single document summarization dataset, XSum (Narayan et al., 2018), a highly abstractive dataset consisting of single-sentence summaries.

### F.1  RQ1 RESULTS: CAN SP-SEARCH REPRESENT THE SUMMARIZATION PROCESS?

As part of answering RQ1, we keep the paraphrase and compression modules unaltered and only retrain the fusion module for XSum, with the training data obtained using the same heuristics as used for CNN/DM (Lebanoff et al., 2019). We also use the same hyperparameters for SP-SEARCH as those used for CNN/DM and compare all methods on 1000 randomly chosen validation samples of XSum. As shown in Table 10, SP-SEARCH obtains an R-2 of 29.60 and an R-L of 48.48, outperforming all baseline methods by a significant margin. Unlike CNN/DM, extractive baselines like 'Lead 1', 'SP-SEARCH Leaves', and 'Top-1 Sentence' do not perform well for XSum, while the abstractive baseline, BART-Oracle does significantly better. The SP-SEARCH results are also lower than in CNN/DM because of the highly abstractive nature of the dataset and the relative difficulty in emulating reference summaries.

### F.2  RQ2 RESULTS: EVALUATION OF SUMMARIZATION PROGRAM GENERATION MODELS

Next, for RQ2, we train a Joint SP generation model that generates the program directly from the document.[9] In Table 11, we compare our model with BART and PEGASUS and a random SP generation baseline. While our model outperforms the random SP baseline, there's significant room for improvement compared to state-of-the-art methods, as indicated by the 16 point gap in R-2 compared to the SP-SEARCH oracle.

## G  EXAMPLES OF SUMMARIZATION PROGRAMS

Figures 3, 4, 5, and 6 show some examples of Summarization Programs identified by SP-SEARCH for human-written summaries. Figures 7, 8, and 9 show examples of Summarization Programs and corresponding summaries generated by our Extract-and-Build SP generation model. For compactness, we show the Summarization Programs by merging the common nodes between the trees.

---

[9] We do not experiment with the *Extract-and-Build* model because extractive models do not perform well on XSum.

|  | R1 | R2 | RL | RLsum |
|---|---|---|---|---|
| BART (Lewis et al., 2020) | 45.14 | 22.27 | 37.25 | 37.25 |
| PEGASUS (Zhang et al., 2020) | 47.21 | 24.56 | 39.25 | 39.25 |
| Random SP (Joint) | 28.71 | 9.20 | 22.43 | 22.43 |
| SP Model (Joint) | 33.56 | 12.18 | 26.18 | 26.17 |
| SP-SEARCH (Oracle) | 51.78 | 28.88 | 48.29 | 48.29 |

**Table 11:** RQ2 – Comparison of our Joint SP generation model with state-of-the-art abstractive summarization models on the XSum test set.

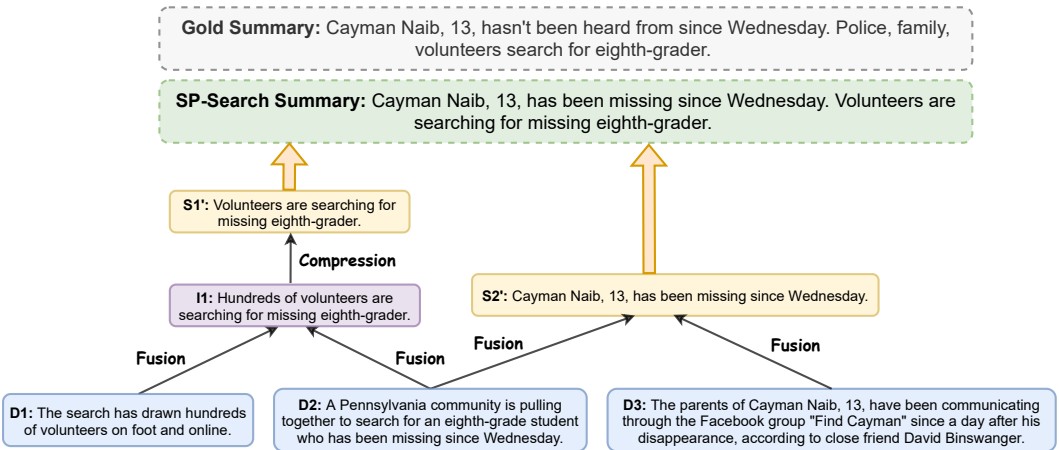

**Figure 3:** Example of a Summarization Program identified by SP-SEARCH. The summary identified by SP-SEARCH matches closely with the gold summary.

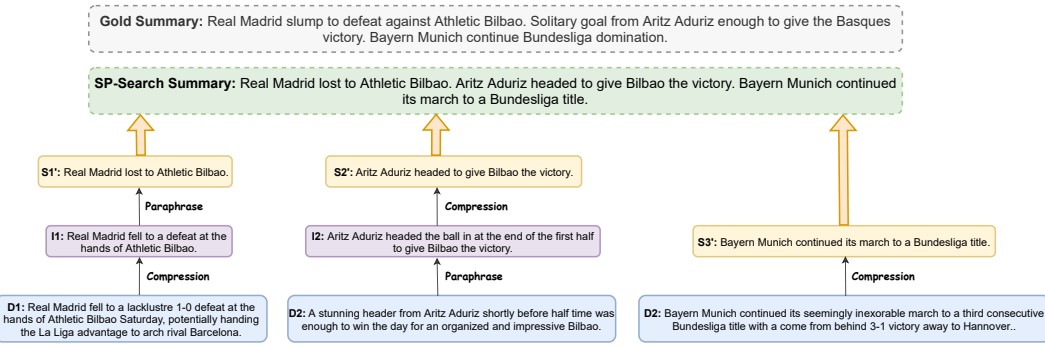

**Figure 4:** Example of a Summarization Program identified by SP-SEARCH. The summary identified by SP-SEARCH matches closely with the gold summary.

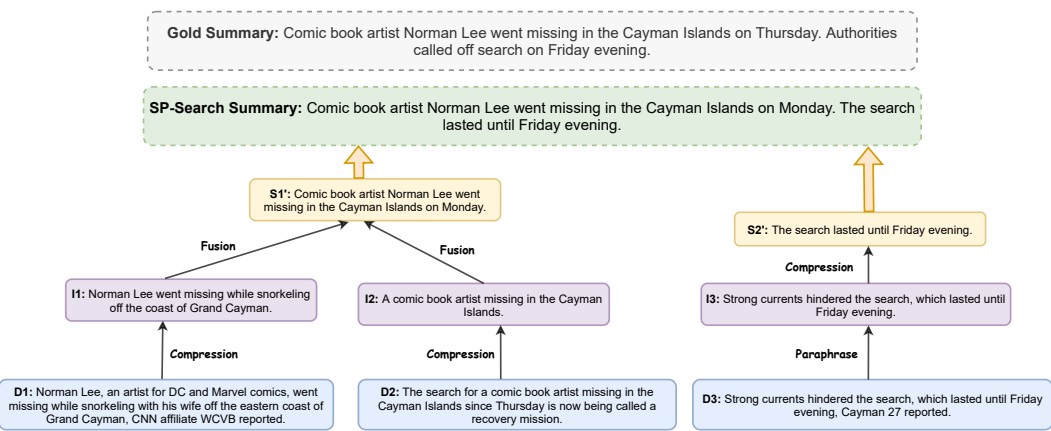

**Figure 5:** Example of a Summarization Program identified by SP-SEARCH. The summary identified by SP-SEARCH contains a factually incorrect part "on Monday". The Summarization Program lets us identify the exact step in which the error was introduced.

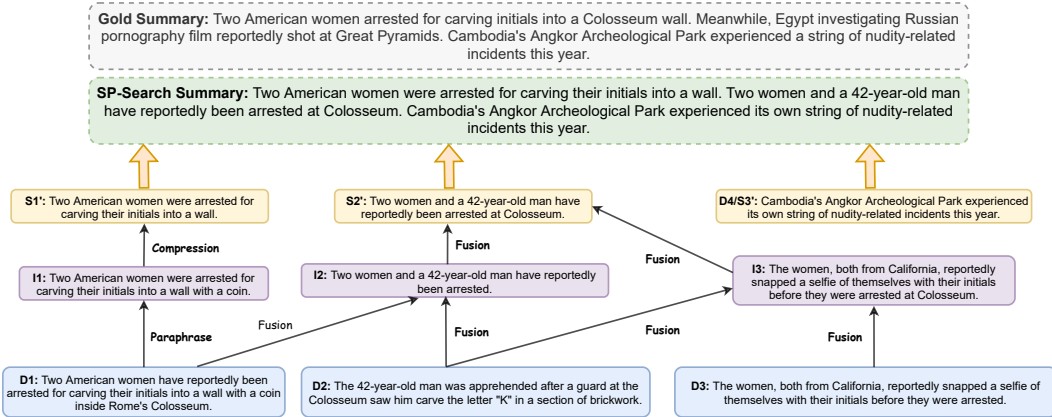

**Figure 6:** Example of a Summarization Program identified by SP-SEARCH. It contains a singleton node (D4/S3). The first two summary sentences have overlap in information, originating from the independence assumption in the generative process of each summary sentence.

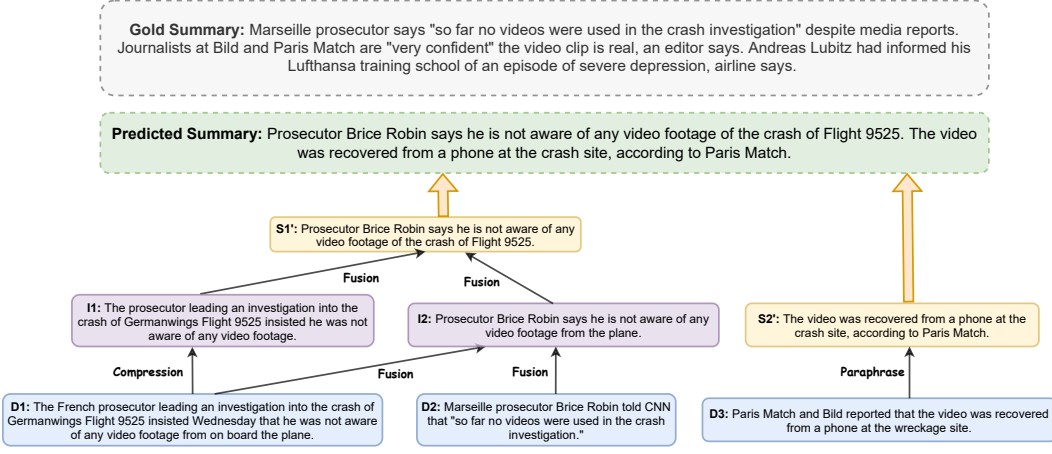

**Figure 7:** Example of a Summarization Program and summary generated by the Extract-and-Build SP generation model.

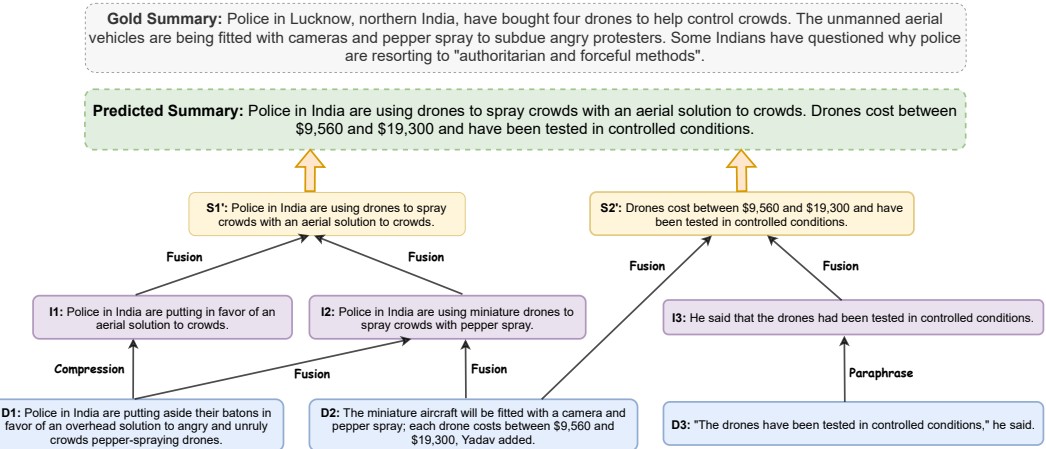

**Figure 8:** Example of a Summarization Program and summary generated by the Extract-and-Build SP generation model.

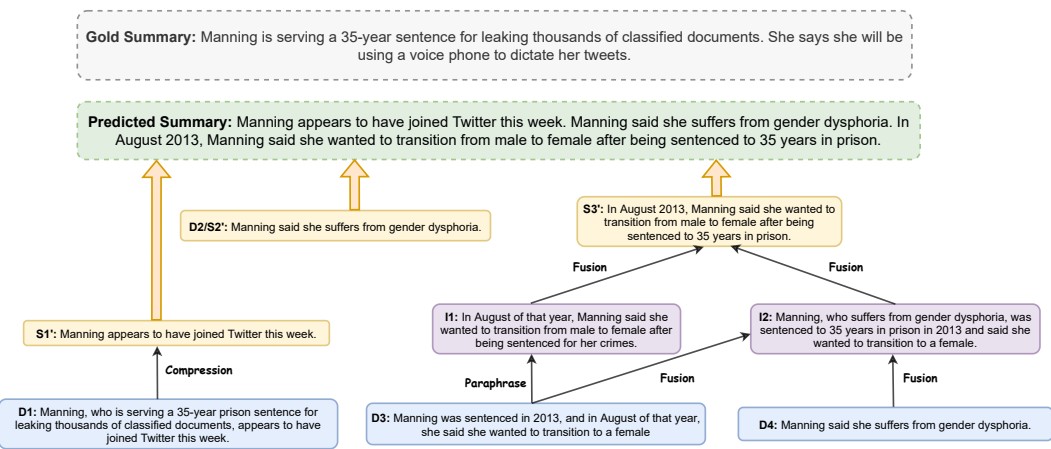

**Figure 9:** Example of a Summarization Program and summary generated by the Extract-and-Build SP generation model. The generated summary contains a factual error that Manning was sentenced to "35 years in prison".

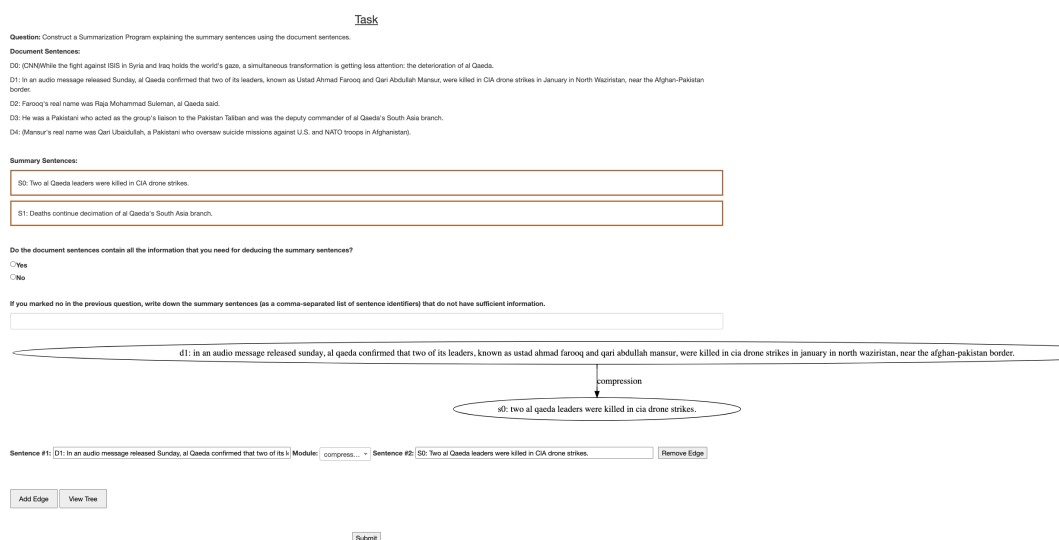

**Figure 10:** Interface for simulating Summarization Programs given a source document and a model summary. Annotators construct trees for all summary sentences by creating one edge at a time.

# H SIMULATABILITY INTERFACE OF SUMMARIZATION PROGRAMS

In Figure 10, we show the interface for our simulatability study in which annotators are asked to write Summarization Pograms for model-generated summaries.

