# OpenReview forum: "Summarization Programs: Interpretable Abstractive Summarization with Neural Modular Trees"
_ICLR.cc/2023/Conference — ICLR 2023 poster_

### Official Review · Reviewer_G3xd · 2022-10-24

**Confidence:** 4
**Correctness:** 4
**Technical Novelty And Significance:** 4
**Empirical Novelty And Significance:** 3
**Recommendation:** 6

**Clarity, Quality, Novelty And Reproducibility:**

Clarity: extremely clear. The paper proposes a non-trivial pipeline, but is detailed without loosing the big picture
Quality: high
Novelty: very original
Reproducible: I would rate this low, accompanying source code. There are lots of small design choices which have been taken - and while mentioned - I would expect those to be hard to reproduce

Minor comments (not sure where to put those):
- authors mention that the paraphrase module also performs compression. This does not seem surprising, as it starts from a summarization model (Pegasus)
- on “When none of the top-k outputs is well-formed (<1% of the samples), we generate the corresponding extractive summary” ->
what is the corresponding extractive summary for “Joint Summarisation Program Generation Model”, which does not generate an intermediate extractive summary?


**Strength And Weaknesses:**

Strengths

1. This is a very original paper, in an undervalued research line - that of interpretable summarization. The proposal is compelling and its advantages well argued for
2. The empirical results showing the potential of proposed model are promising. The best obtainable summary is +20 Rouge-L points above what can reasonably be expected to be obtained with state-of-the-art methods

Weakness
1. The second set of experiments are underwhelming. Not only do they perform rather far from current pure neural methods, they do so even when using those as starting point: "Extract-and-Build" (RL=21.8) starts from the extractive summary obtained through MatchSum (RL=28.7). A very ruthless reading of Table 3 would be that the proposed technique starts from something good and turns it into something worse, although not as bad as random selection.
The authors discuss why this could be the case, but the results in the paper do not indicate that this gap could be overcome.
2. Those underwhelming results come as a total surprise to the reader, as they are not forshadowed in the abstract or introduction. This disappointment could easily be avoided, at the cost of turning away those readers that only come for the SOTA results (which would in any case not be interested after finishing the paper)
3. The simulation protocol is very interesting to study the intepretability of SP programms, but it doesn't seem to me that too much should be read into it. In short, in the study humans have to predict the program before and after seeing the program explained by the proposed system. The Rouge score of the summaries obtained by the human-chosen program _after_ the training phase is slightly better. However, it should be noted that the 3 humans who performed this were authors of the paper. So, on one side they were clearly not neutral, and on the other hand they probably started with a very good understanding from the form of the program (therefore the difference might be smaller than with non-experts).
4. Experiments are only reported on one dataset, and the last years of summarization papers have but the bar higher than that.


**Summary Of The Paper:**

This paper proposes to model summaries in the form of "summarization program", a sequence of 3 actions (compression, fusion, paraphrase) over sentences. Such an approach has the advantage of being more interpretable, as the final summary comes together with a "proof" on how it was obtained.
This open very interesting avenues for interactive summarization, as well as inspection of the intermediate steps performed by the system.

Two types of experiments are reported in this paper:
1. An exploratory one, which aims to evaluate what could be achieved with such an approach. This is done in by optimizing for the oracle summary.
2. A predictive one, which evaluates the summaries predicted on unseen example (the more traditional ML approach)

**Summary Of The Review:**

An original proposal, with lots of potential for interpretability work. The search-space seems wide enough to also be possible to use the method to obtain SOTA results - as measured by Rouge scores. However, this potential is far from being realized in thie paper.

My recommendation is still for acceptance, because of its originality and because it takes "the path less trodden".

---

> ### Author Response · Authors · 2022-11-18
> **Response to Reviewer G3xd**
>
> We appreciate your time, effort, and detailed comments on our paper. We would also like to thank you for acknowledging the originality of our work. We address your comments and questions below.
>
> > **The second set of experiments are underwhelming. A very ruthless reading of Table 3 would be that the proposed technique starts from something good and turns it into something worse, although not as bad as random selection. The authors discuss why this could be the case, but the results in the paper do not indicate that this gap could be overcome.**
>
> We agree that our models underperform state-of-the-art abstractive summarization models. However, we want to note that our models are promising first steps in this new research direction of generating programs for interpretable abstractive summarization and as you point out, the high oracle scores show that there is plenty of scope for improvement.
>
> Our Extract-and-Build model starts with an extractive model that has high ROUGE scores while our final abstractive model achieves lower scores than that. This is actually not too surprising when we consider that on CNN/DM, SOTA extractive models achieve similar ROUGE scores as SOTA abstractive models. Hence, this might well be a dataset-specific observation and on more abstractive datasets, an Extract-and-Build model might lead to higher ROUGE scores than the extractive summary it starts off with.
>
> Regarding how one may close the gap between the extractive and abstractive ROUGE scores, this may not be essential and progress can be made in several different ways. For example, our second model, Joint SP generates SPs directly from the document by bypassing any extractive stage. It is possible that future research finds this class of models to be a more promising approach toward better SP generation models, in which case the gap between extractive and abstractive summaries in a 2-stage model is of little significance. Another orthogonal direction could be to build the SPs in a step-wise manner (similar to how the search proceeds) but by designing some reference-free scoring functions that evaluate each intermediate step and also decide when to stop growing the trees further. Such scoring functions can be trained on the oracle programs.
>
> > **Those underwhelming results come as a total surprise to the reader, as they are not forshadowed in the abstract or introduction. This disappointment could easily be avoided, at the cost of turning away those readers that only come for the SOTA results (which would in any case not be interested after finishing the paper)**
>
> Thank you for the suggestion, we have updated our introduction accordingly!
>
> > **The simulation protocol is very interesting to study the intepretability of SP programms, but it doesn't seem to me that too much should be read into it.**
>
> As you rightfully point out, we ourselves carried out the simulation study because we understand the task well. For an initial study, we chose to prefer task understanding, reliability, and dedication of the annotators more. We also think that any *bias* would have to be in the direction of underperforming on the pre-test. As long as we didn't accidentally underperform on the pre-test, there shouldn't be bias in the results. And in fact, it's probable that we overperformed on the pre-test relative to what average annotators with no familiarity with the task would do, hence leading to smaller gap / worse results for us than with non-experts. This is a very different kind of bias than doing, for example, an A vs. B summary comparison where we know which summary came from which system.

---

> > ### Author Response · Authors · 2022-11-18
> > **Response to Reviewer G3xd (continued)**
> >
> > > **Experiments are only reported on one dataset, and the last years of summarization papers have but the bar higher than that.**
> >
> > We would like to note that our work is different from other recent summarization papers that focus on improving summarization performance on existing datasets. In that regard, our choice of a dataset, CNN/DM, is a case study for this new paradigm of Summarization Programs. That said, following your suggestion, we added new experiments with XSum for both of our research questions in Appendix E. Here is a brief summary of the results.
> >
> > For RQ1, we observe that SP-Search obtains significant improvements over other baselines, achieving R-2 of 30. The extractive baselines do not work well on this dataset because XSum contains highly abstractive one-sentence summaries. Our search results are also lower than in CNN/DM because of the same reason and the relative difficulty in emulating reference summaries.
> >
> > |                  |     R1     |     R2    |     RL    |   RLSum   |
> > |------------------|:----------:|:---------:|:---------:|:---------:|
> > | Lead-1           |    15.88   |    1.54   |   11.71   |   11.81   |
> > | SP-Search Leaves |    25.10   |    5.89   |   16.48   |   19.44   |
> > | Top-1 Sentence   |    29.77   |    7.21   |   20.85   |   21.13   |
> > | Random SP        |    28.96   |    9.04   |   22.38   |   22.41   |
> > | BART-Oracle (Sample)        |    49.23   |    25.82   |   42.31   |   42.30   |
> > | SP-Search Top-1        |    47.72  | 24.83 | 43.50 | 43.50
> > | SP-Search        |    **51.86** | **29.60** | **48.48** | **48.51**
> >
> > For RQ2, we train a Joint SP model, which obtains an R-L of 12. The gap with state-of-the-art models is significant but the oracle results are again encouraging for future work.
> >
> > |                  |     R1     |     R2    |     RL    |   RLSum   |
> > |------------------|:----------:|:---------:|:---------:|:---------:|
> > BART | 45.14 | 22.27 | 37.25 | 37.25
> > PEGASUS | 47.21 | 24.56 | 39.25 | 39.25
> > Random SP (Joint) | 28.71 | 9.20 | 22.43 | 22.43
> > SP Model (Joint) | 33.56 | 12.18 | 26.18 | 26.17
> > SP-Search (Oracle) | 51.78 | 28.88 | 48.29 | 48.29
> >
> > > **Reproducible: I would rate this low, accompanying source code. There are lots of small design choices which have been taken - and while mentioned - I would expect those to be hard to reproduce**
> >
> > We have already attached our source code and upon publication, we will release it publicly so that it helps with the reproducibility. We will also make sure to release all our SP-Search programs for both CNN/DM and XSum to enable building better models, as part of future research.
> >
> > > **Authors mention that the paraphrase module also performs compression. This does not seem surprising, as it starts from a summarization model (Pegasus)**
> >
> > We agree with you! In an ideal world, it is probably worthwhile trying to make the modules totally complementary but in practice, it might still be hard to achieve. We hope that future work can look into this more closely.
> >
> > > **On “When none of the top-k outputs is well-formed (<1% of the samples), we generate the corresponding extractive summary” -> what is the corresponding extractive summary for “Joint Summarisation Program Generation Model”, which does not generate an intermediate extractive summary?**
> >
> > Thanks for pointing this out! In the Joint SP model, we extract the sentence identifiers from the generated SP string and concatenate the corresponding sentences to form the extractive summary. For example, if the model generates an ill-formed output like  “[ fusion ( <D0> , <D1> ) ) ] [ paraphrase ( <D4> )  ]”, we construct “ [ <D0> ] [ <D1> ] [ <D4> ]” as the extractive summary. We have clarified this in the updated version (last sentence of Section 5).

---

> > > ### Comment · Reviewer_G3xd · 2022-11-25
> > > **Thanks & on additional datasets**
> > >
> > > Thanks for those clarifications and additional comments - I hope they will strengthen the paper.
> > > CNN/DailyMail and XSum are known to be very noisy (on top of the usual news-article position bias). A recent paper which re-discovered this is https://arxiv.org/abs/2211.09110
> > >
> > > If you do more experiments, I would recommend using more reliable test sets.

---

> > > > ### Author Response · Authors · 2022-12-08
> > > > **Response to additional comments by Reviewer G3xd**
> > > >
> > > > We thank the reviewer for their additional comments and we do agree that their suggestions and our new results indeed make the paper stronger. Following their suggestion, we carried out some more experiments on Wikihow, a non-news domain, potentially higher quality dataset. Our conclusion is that SP-Search is quite generic and works very well for a range of summarization datasets (again with no hyperparameter tuning). Wikihow is still a very abstractive dataset, and thus the oracle results aren't as high as CNN/DM, but nevertheless, the RL scores are considerably better.
> > > >
> > > > |                                 | R1    | R2    | RL |
> > > > |---------------------------------|-------|-------|-------|
> > > > | PEGASUS        | 43.06 | 19.71 | 34.80 |
> > > > | SP-Search | 49.49 | 23.53 | 48.59 |

---

### Official Review · Reviewer_wpG8 · 2022-10-25

**Confidence:** 4
**Correctness:** 3
**Technical Novelty And Significance:** 2
**Empirical Novelty And Significance:** 2
**Recommendation:** 3

**Clarity, Quality, Novelty And Reproducibility:**

‒ Overall the description is clear and easy to follow, with a good illustration of SPs

‒ Authors explain evaluating Rouge-L during SP-SEARCH leads to better performance, but it is unclear why Rouge-1 is used to first select the top-k document sentences

‒ It is unclear, for the first research question, whether the BART-Oracle is fine-tuned on the training data, whether the score for beam search is an average of the top-10 summaries or the best of all  10 summaries.

‒ Although the idea is quite new, there is a lack of novelty in terms of the models used.

‒ The paper is likely to be reproducible as the experimental settings are reported in detail.


**Strength And Weaknesses:**

Strength:
‒ SP is an interesting and novel idea

‒ The design makes it much easier to trace the generation process, and to manually detect if any hallucination has occurred; the method helps to limit the issue of hallucinations down to mostly a single module (fusion)

Weaknesses:

On experiments for RQ2

‒ The results for generating SPs are much weaker than the SOTA both in terms of fluency and factuality: as compared to the model for the base architecture (BART), the best SP model experience a 6.53 drop in Rouge-1 and a 2.27 drop in QuestEval. Even though the authors show the Oracle scores to be much higher, it is not guaranteed if such potential can be realized since the Oracles make use of gold in the first place.

‒ From what I understand, for the RQ2 experiments, authors only finetuned BART for generating SPs (or used an existing model for the sentence extraction stage). There is little novelty and the approach seems unnatural. It is possible that the target output (SPs) differs too much from the pre-trained text. The three modules (fusion, compression, and paraphrase) are decoupled from the SP-generating module, and it is possible that the SP-generating module is totally unaware what are the exact implications of its generated modular instructions.

‒ The authors used three separate large neural models for paragraph, compression, and fusion; from the perspective of parameter sizes, it may be unfair to compare the performance of the SP Models to the BART or PEGASUS models.

On experiments for RQ1

‒ It is not explicitly stated in the paper, but from what I understand, the BART-Oracles (Beam/Sample) are first fine-tuned on the training set and then used for generation? In this case, it is not surprising that it should have worse results than SP-SEARCH, which refers to the gold summaries for practically every step of its search process.

**Summary Of The Paper:**

This paper proposes a new framework, summarization programs (SPs), for interpretable summarization. It starts with source document sentences as leaves, applies a series of transformations of fusion, compression, and paraphrasing, and finally leads to the root nodes for each summary sentence. Such design makes the generation process clear to humans and makes it easier to catch any hallucination created along the way. The authors come up with a best-first search algorithm, SP-SEARCH, to take document sentences and gold summaries as input, and output SPs for supervised training. They also attempt to train models to generate the SPs, and conduct simulation studies to gauge the interpretability of the summarization. Nevertheless, their results still lag behind those of the SOTA models.

**Summary Of The Review:**

Overall this paper proposes an interesting idea, and the writing is easy to follow. Most details on the experiments are reported clearly, and there is extensive analysis of the results as well. However, there is limited novelty in the model, and the proposed approach is too simple and cannot give good results as compared to the SOTA models.

---

> ### Author Response · Authors · 2022-11-18
> **Response to Reviewer wpG8**
>
> We appreciate your time, effort, and detailed comments on our paper. We would also like to thank you for appreciating our idea and extensive analysis of the results. Please find the answers to your comments and questions below:
>
> > **The results for generating SPs are much weaker than the SOTA both in terms of fluency and factuality: as compared to the model for the base architecture (BART), the best SP model experience a 6.53 drop in Rouge-1 and a 2.27 drop in QuestEval. Even though the authors show the Oracle scores to be much higher, it is not guaranteed if such potential can be realized since the Oracles make use of gold in the first place.**
>
> It’s true that our models underperform state-of-the-art summarization models but we lay out the first such promising setup for Summarization Programs with these 3 intuitive modules that show that the oracle results are high enough for future work to build on top of our methods (as also noted by Reviewer G3xd).
>
> Second, we’d like to clarify that our focus is on *interpretability in abstractive summarization* and not to develop a state-of-the-art model. In that regard, we (1) proposed Summarization Programs, a novel paradigm for explaining abstractive summarization that all reviewers agree is “exciting” and “original”, (2) developed a novel search algorithm that can effectively decode the step-wise generative process of human summaries, and (3) introduced some initial models for generating Summarization Programs, which although not the best, provide strong starting points for better models in the future.
>
> Third, interpretability-accuracy tradeoff is common and past studies have also made similar observations [1, 2].
>
> Finally, we list some potential directions that could build on top of our work. (1) Our models generate the entire program end-to-end in one go. Another approach could be to build it in a step-wise manner (similar to how the search proceeds) but by designing some scoring functions that evaluate each intermediate step and also decide when to stop growing the trees further. Such scoring functions can be trained on the oracle programs. (2) The two strategies of end-to-end vs step-wise program generation can also be combined. For example, in the first stage, an end-to-end model outputs an initial program, and then, a search-like step-wise process performs further exploration on top of it to generate better programs. In summary, we see a lot of scope for future work and we hope that our initial models can encourage the community to make further progress in the task of interpretable abstractive summarization.
>
> [1] Interpretable Neural Predictions with Differentiable Binary Variables. Jasmijn Bastings, Wilker Aziz, Ivan Titov. ACL 2019
>
> [2] Learning to Faithfully Rationalize by Construction. Sarthak Jain, Sarah Wiegreffe, Yuval Pinter, Byron C. Wallace. ACL 2020
>
> > **From what I understand, for the RQ2 experiments, authors only finetuned BART for generating SPs (or used an existing model for the sentence extraction stage). There is little novelty and the approach seems unnatural.**
>
> There has indeed been past work on generating structured outputs or programs using seq2seq models. However, we want to emphasize that our novelty is in taking a program approach for the abstractive summarization task, which is also one of the harder tasks for developing such an approach. We also respectfully disagree that our approach is unnatural because seq2seq models have been extensively applied and shown to work well for generating structured outputs like parses, proofs, etc [1, 2, 3].
>
> [1] Neural AMR: Sequence-to-Sequence Models for Parsing and Generation. Ioannis Konstas, Srinivasan Iyer, Mark Yatskar, Yejin Choi, Luke Zettlemoyer. ACL 2017
>
> [2] ProofWriter: Generating Implications, Proofs, and Abductive Statements over Natural Language. Oyvind Tafjord, Bhavana Dalvi Mishra, Peter Clark. Findings of ACL 2021
>
> [3] Seq2seq Dependency Parsing. Zuchao Li, Jiaxun Cai, Shexia He, Hai Zhao. COLING 2018
>
> > **The authors used three separate large neural models for paragraph, compression, and fusion; from the perspective of parameter sizes, it may be unfair to compare the performance of the SP Models to the BART or PEGASUS models.**
>
> Yes, we make use of three large models for the modules but please note that the programs are still generated by only a single BART model. Hence, as far as the parameter space and training time of the program generation models are concerned, it’s a fair comparison. During inference, when the programs are parsed to generate summaries, our model may use more parameters (based on which modules are activated) but we believe that such trade-offs are worth it for achieving interpretability.

---

> > ### Author Response · Authors · 2022-11-18
> > **Response to Reviewer wpG8 (continued)**
> >
> > > **It is not explicitly stated in the paper, but from what I understand, the BART-Oracles (Beam/Sample) are first fine-tuned on the training set and then used for generation? In this case, it is not surprising that it should have worse results than SP-SEARCH, which refers to the gold summaries for practically every step of its search process.**
> >
> > Yes, we first finetune a BART model on the CNN/DM training data. From that fine-tuned model, we sample top-10 summaries and report the scores for the *best* summary that has the *highest* ROUGE-L score with the gold summary. We call this model Oracle because it compares multiple summaries with the gold summary and chooses the best one. It is true that the BART-Oracle model uses the reference summary only once (for the final generation) whereas SP-Search uses the reference summary at every step. This is the exact point of this experiment which is to show that step-wise structured decoding (SP-Search) leads to a closer emulation of the reference summary compared to generating a summary in one go and then comparing the final summary (BART-Oracle).
> >
> > > **Authors explain evaluating Rouge-L during SP-SEARCH leads to better performance, but it is unclear why Rouge-1 is used to first select the top-k document sentences.**
> >
> > Thanks for the interesting question! We did conduct this experiment in the past and found that our search algorithm is fairly robust to what ROUGE metric is used to select the top-k initial sentences (IS). As shown below, R-2 does slightly better than R-1 and R-L but overall, we still obtain sufficiently high oracle results of 40-41 R-2 through SP-Search, irrespective of the selection metric. The top-k sentences provide a small candidate set to build the Summarization Programs on. Hence, we chose to use a more surface-level metric like ROUGE-1 instead of stricter metrics like ROUGE-L. We have updated the paper with this new result in Appendix C and Table 8.
> >
> > | IS Metric |     R1    |     R2    |     RL    |   RLSum   |
> > |-----------|:---------:|:---------:|:---------:|:---------:|
> > | R1        | 61.88     | 40.11     | 58.46     | 60.32     |
> > | R2        | **62.07** | **41.61** | **59.05** | **60.76** |
> > | RL        | 61.79     | 40.36     | 58.66     | 60.38     |

---

> > > ### Author Response · Authors · 2022-11-29
> > > **Response to Reviewer wpG8**
> > >
> > > Dear reviewer, this is a gentle reminder for our response above to your questions. We would very much appreciate it if you could let us know whether we have answered all of your questions. If there are additional questions or clarifications, please let us know and we will be happy to answer those as well. If you are satisfied with our responses, please consider updating your scores. Thanks again!

---

> ### Comment · Reviewer_wpG8 · 2022-12-01
> **Additional comments on Dec 1st.**
>
> Thanks for the authors' ​detailed response and sorry for responding late since it took a bit longer time to redigest your paper combined with your response.
>
> My question on why Rouge-1 is used to select the top-k document sentences is completely addressed. Nevertheless, I still have some remaining concerns.
>
>
> For the results on CNN/DM:
>
> ● I understand the authors may want to focus on interpretability, nevertheless, the drop from SOTA is significant (as also noted by Reviewer G3xd). My concern is that even if the interpretability is good if the performance gap is too large, there may be limited application values. The authors explained that the oracle results are high which indicates good potential. I am still not convinced, because the comparison is made with the Oracle BART, which is BART generations and not a leveled comparison. This lead to my second point below.
>
> ● The authors compare the best-selected summary generated from BART with their SP-computed summary and conclude that SP has better potential. However, the abstractive or extractive models are trained directly using the reference summary, whereas the authors' models are trained using the SP-computed summary. Comparing Bart-generated gold with SP-computed summary is not fair and with all respect,  I disagree that the latter being higher would indicate good performance potential. Rather, we should compare the SP-computed summary with the reference summary.
>
> ● Authors provided some future directions, nevertheless these still didn't link directly to how any of these methods can lead to better performance. Even if the main focus of this paper is interpretability, my stand is still that the performance gap shall not be too large, as we need the model to perform reasonably in the first place, before any additional value can be gained from interpretability.
>
>
> For the modules:
>
> ● One of my concerns remains is that "The three modules (fusion, compression, and paraphrase) are decoupled from the SP-generating module, and it is possible that the SP-generating module is totally unaware of what are the exact implications of its generated modular instructions."
> Maybe a bit more investigation and analysis into the SP model would be good.
>
> ● For my question on the unfair comparison between a single BART model and the three sub-modules used by SP, the authors explain that the underlying three modules are still BART,  and the SP-generating model is comparable to BART. I don't deny such facts, but in fact, in order to generate the summary using the authors’ method, one must first use an SP-model to generate the SP, then according to the SP, selectively use any of the 3 sub BART models, which all holds different parameters (as I understand, they are finetuned on different tasks). As compared to loading the parameters from a single BART model, the authors' approach really requires 4 sets of parameters. From a practical point of view, in order to use this method, one must train 4 models, which is complicated (not to mention deriving the SP references in the first place).
>
>
> Additional Comments:
>
> ● Author experimented on one additional data, XSum, which I do appreciate. Does it seem that the SP benefits the dataset with a short target length more? XSum has single-sentence summaries, whereas CNN/DM contains only a few sentences in its reference summaries. As the generation increases, the complexity of generating the SP also increases, and another interesting future direction may be how to enhance interpretability for long generations or to optimize the SP-search algorithm.
>
> ● From what I understand, for the RQ2 experiments, authors only finetuned BART for generating SPs (or used an existing model for the sentence extraction stage). There is little novelty and the approach seems unnatural. It is possible that the target output (SPs) differs too much from the pre-trained text. The three modules (fusion, compression, and paraphrase) are decoupled from the SP-generating module, and it is possible that the SP-generating module is totally unaware what are the exact implications of its generated modular instructions.
>
>
>
> Therefore, given these unclear aspects and not-so-practical settings, I would insist on my original recommendation.

---

> > ### Author Response · Authors · 2022-12-08
> > **Response to additional comments by Reviewer wpG8**
> >
> > We thank the reviewer for their additional comments. However, we think that there are still some misunderstandings which we clarify below.
> >
> > > **The authors explained that the oracle results are high which indicates good potential. I am still not convinced, because the comparison is made with the Oracle BART, which is BART generations and not a leveled comparison.**
> >
> > Our paper tackles two research questions that answer two very different aspects of summarization programs. Please note that Oracle-BART is a baseline from the first RQ. It was compared to the SP-Search summary for a very different purpose, as discussed below, and has got nothing to do with the fact that SP models have good potential.
> >
> > The first research question answers that if one has access to a human-written summary, what is the best one can do to emulate the generative process of that summary? In that regard, the comparison with Oracle-BART shows that step-wise decoding with the three modules leads to a better representation of human summaries. The fact that the SP models have good potential comes from the observation that our SP-Search algorithm achieves high rouge scores and hence, a model trained on such data has the potential to obtain ROUGE scores similar to that of SP-Search. This is independent of the scores BART-Oracle obtains for RQ1.
> >
> > > **Comparing Bart-generated gold with SP-computed summary is not fair and with all respect, I disagree that the latter being higher would indicate good performance potential. Rather, we should compare the SP-computed summary with the reference summary.**
> >
> > Again, we think that there’s some misunderstanding here. The SP-computed summary is indeed compared with the reference summary and that is how we obtain ROUGE-2 scores of 40. As we mentioned in our last answer, Research Question 2 has nothing to do with the BART-oracle summaries (if that’s what the reviewer means by Bart-generated gold summaries). Hence, the only pertinent comparisons for SP models are state-of-the-art methods like BART and PEGASUS. Now, our SP models do demonstrate lower performance than BART and PEGASUS and there are two factors worth noting here.
> >
> > * First, BART and PEGASUS optimize for the gold summaries while our SP models optimize for the SP-Search summaries. SP-Search, although does a good job of emulating human summaries, still isn’t perfect. So, one way to improve SP models is to train them on even better SP-Search oracle summaries. This can be done by incorporating more modules or further enhancing the existing ones by making them more fine-grained, etc (as Reviewer T8vS suggested in their review).
> > * Second, even if one does not improve the upper bound / oracle summaries further, there’s still a large 24-point gap in R-2 that can be closed by just training on our current SP-Search summaries. While it may be unrealistic to develop an SP model that matches SP-Search, it is definitely not unrealistic to be able to match SOTA models like BART (which is only 5 points better off). We listed some potential approaches in our previous response like generating the programs step-wise similar to how the search operates.

---

> > > ### Author Response · Authors · 2022-12-08
> > > **Response to additional comments by Reviewer wpG8 (continued)**
> > >
> > > > **One of my concerns remains is that "The three modules (fusion, compression, and paraphrase) are decoupled from the SP-generating module, and it is possible that the SP-generating module is totally unaware of what are the exact implications of its generated modular instructions." Maybe a bit more investigation and analysis into the SP model would be good.**
> > >
> > > The job of the SP model is to learn meaningful patterns of which sentences can be fused or which sentence can be compressed, etc. So, even if the model does not understand the exact semantics of what a fusion module is, as long as it learns to associate useful features with these modules, we think that this in itself can lead to a reasonable SP model. That’s why when training the model, we provide sentence identifiers on the input side. For instance, if the model picks up signals like the "FUSION" keyword comes up between two sentence identifiers $S_i$ and $S_j$ whenever there is some overlap between the two sentences, it is implicitly learning fusion.
> > >
> > > To investigate the effectiveness of our SP model further, we conducted two experiments. First, we asked that if SP-Search were to fully emulate the human summaries, i.e., the target summaries are SP-Search summaries, how would our current SP model perform (see table below)? Encouragingly, we find that our SP model’s summaries, when evaluated against SP-Search summaries, obtain R-2 scores of 21, which is as good as BART summaries that are evaluated against reference summaries. This shows two things: (1) the SP model is an effective model, given the (SP-Search) summaries it is optimizing for, and (2) even if one does not improve upon our SP model further but only focuses on improving SP-Search and retrains the model on that new data, that in itself can help close the gap to SOTA models.
> > >
> > > |                                 | R1    | R2    | RL    | RLSum |
> > > |---------------------------------|-------|-------|-------|-------|
> > > | BART (eval w/ Ref Summary)           | 44.19 | 21.29 | 31.06 | 41.04 |
> > > | SP Model (eval w/ Ref Summary)       | 37.23 | 16.40 | 26.48 | 34.18 |
> > > | SP Model (eval w/ SP-Search Summary) | 39.76 | 21.63 | 30.33 | 37.37 |
> > >
> > > Second, we analyzed the distribution of the tree structures learned by our SP model. We observe that it models the training data well and generates tree structures that are distributed very similarly to those shown in Table 5 of the paper. We hope that both these experiments demonstrate that our SP models provide simple and effective ways of learning SPs from source documents.
> > >
> > > > **Author experimented on one additional data, XSum, which I do appreciate. Does it seem that the SP benefits the dataset with a short target length more?**
> > >
> > > Thank you for appreciating our additional experiments on XSum. XSum is a more challenging dataset than CNN/DM because the summaries are highly abstractive. Nevertheless, SP-Search still obtains high enough oracle scores, which shows that the three intuitive operations of paraphrase, compression, and fusion can also decode highly abstractive summaries fairly well.
> > >
> > > > **From what I understand, for the RQ2 experiments, authors only finetuned BART for generating SPs (or used an existing model for the sentence extraction stage). There is little novelty and the approach seems unnatural.**
> > >
> > > We were wondering if the reviewer was able to see our earlier response [here](https://openreview.net/forum?id=ooxDOe7ZtBe&noteId=S3BnzN9vWB) where we address this specific concern by citing multiple related papers. In summary, it doesn't matter that program generation is not next-token prediction, pretrained LMs do well at program generation when finetuned (please refer to the references in our previous response). It's common in NLP that models with next token prediction are finetuned for various purposes and we respectfully disagree that it's "unnatural".
> > >
> > > In the context of summarization, an SP can be thought of as a plan. As humans, even before writing out a summary, we come up with a plan that conveys, for instance, "compress sentence 1, fuse sentences 2 and 3, etc". The exact outputs of these operations are determined at a later stage when the plan is executed. Hence, our method is neither unnatural from a modeling perspective nor from the perspective of how humans summarize documents.

---

### Official Review · Reviewer_T8vS · 2022-10-25

**Confidence:** 4
**Correctness:** 3
**Technical Novelty And Significance:** 4
**Empirical Novelty And Significance:** 4
**Recommendation:** 8

**Clarity, Quality, Novelty And Reproducibility:**

The analysis of search could be more visualized and comprehensive. Some interesting baselines to see: an always-right-branched search tree, or a binary fusion tree like fusion(fusion(0,1), fusion(2,3)). I am not suggesting that you need to do these during the rebuttal. The concern I have in mind is: what’s the added value of the search program/function, and how hard to get it work. If there is no easy recipe to tune {top-k, filtering, PQ, height, etc.} on a new dataset, it’s hard to build real systems upon this paper.




Two missing references:

Learning to Fuse Sentences with Transformers for Summarization
Lebanoff et al.

Single Document Summarization as Tree Induction
Liu et al.

**Strength And Weaknesses:**

Accept:
The paper introduces the notion of “program” to decompose the generation of summaries. It’s an exciting direction that blurs the boundary of extractive and abstractive, end-to-end and modular summarization systems.

The paper carefully chooses components and explains the reason behind the choices.

Reject:
There are some reasons to “reject” this paper, but they are mostly inevitable on the way building a complicated modular systems.

The search algorithm and process is very complicated, and I intuitively can imagine how fragile it is.
The experimental results (non-oracle) are not competitive.

The definition and derivation of the three meta operations are worth further consideration. For example, compression and paraphrasing are highly overlapped with each other. The granularity of operation can also be more fine-grained (EDUs, clauses, etc.).


**Summary Of The Paper:**

This paper describes a system/paradigm called “summarization programs”. The goal is to build an interpretable search-then-assemble program for abstractive summarization. Best-first search is deployed to find the the neural module operations, including compression, fusion, and paraphrasing. There are comprehensive analysis and results about the proposed model and its ablated variants.
Although the system does not beat end-to-end SOTA models, it provides a data point for building modular text generation systems. The idea behind is novel and exciting, and the author explains the design choices well. I would like to accept this paper.


**Summary Of The Review:**

It's a strong paper with novel ideas and solid experiments. Although the system does not beat end-to-end SOTA models, it provides a data point for building modular text generation systems. The idea behind is novel and exciting, and the author explains the design choices well. I would like to accept this paper.

---

> ### Author Response · Authors · 2022-11-18
> **Response to Reviewer T8vS**
>
> We appreciate your time, effort, and detailed comments on our paper. We are glad that you found our work novel and exciting. We address your comments and questions below.
>
> > **The search algorithm and process is very complicated, and I intuitively can imagine how fragile it is. If there is no easy recipe to tune {top-k, filtering, PQ, height, etc.} on a new dataset, it’s hard to build real systems upon this paper.**
>
> In general, we believe that tuning these hyperparameters aren’t too critical for obtaining good oracle scores. They can obviously provide some gains but as the ablations in Table 6 (in the original version) for CNN/DM suggest, most aspects like the top-k sentences or queue size do not have much impact beyond a certain point. To test this further, we followed your suggestion and evaluated SP-Search on the XSum dataset. This time, we only re-train the fusion model for XSum, with the training data obtained using the same heuristics as used for CNN/DM (Lebanoff et al., 2019). The other modules and the set of search hyperparameters are kept exactly the same as CNN/DM to test the generalizability of our algorithm. It is important to note that XSum is a highly abstractive dataset with one-sentence summaries and where extractive models do not perform well, as shown below. For example, the Top-1 sentence only achieves an R-L of 20.85.
>
> |                  |     R1     |     R2    |     RL    |   RLSum   |
> |------------------|:----------:|:---------:|:---------:|:---------:|
> | Lead-1           |    15.88   |    1.54   |   11.71   |   11.81   |
> | SP-Search Leaves |    25.10   |    5.89   |   16.48   |   19.44   |
> | Top-1 Sentence   |    29.77   |    7.21   |   20.85   |   21.13   |
> | Random SP        |    28.96   |    9.04   |   22.38   |   22.41   |
>
> Encouragingly, SP-Search obtains good oracle scores of R-L 48.29 without any hyperparameter tuning, which is 9 points better than state-of-the-art models on the dataset. We hope that these results are good indicators of the robustness of our algorithm on new datasets. More details of our experiments on XSum can be found in Appendix E of our updated version.
>
> |           |     R1    |     R2    |     RL    |   RLSum   |
> |-----------|:---------:|:---------:|:---------:|:---------:|
> | BART      | 45.14     | 22.27     | 37.25     | 37.25     |
> | PEGASUS   | 47.21     | 24.56     | 39.25     | 39.25     |
> | SP-Search | **51.78** | **28.88** | **48.29** | **48.29** |
>
> > **The definition and derivation of the three meta operations are worth further consideration. For example, compression and paraphrasing are highly overlapped with each other. The granularity of operation can also be more fine-grained (EDUs, clauses, etc.).**
>
> We agree with you that these are interesting directions to consider for future work. In the paper, discussion section 7 talks about some of these. Our work lays out the first such promising setup for Summarization Programs with these 3 intuitive modules but our framework can naturally extend to incorporate more fine-grained modules, and we hope that future work builds on top of our methods to further refine the modules and the granularity of the units/arguments to the modules.
>
> > **What’s the added value of the search program/function?**
>
> There is significant value in being able to search for good oracle programs. We show this through Table 1, in which a random SP with all three modules only achieves R-2 of 15, compared to R-2 of 40 with searched SPs. With respect to constraining the module space, we have ablations in Table 1 showing the utility of each module. Moreover, as per your suggestion, we also added the experiment with “binary fusion trees”. If we understood this correctly, this is equivalent to performing the search with just the fusion module – this model obtains an R-2 of 34, 6 points worse off than our best result. In summary, good oracle programs require both searching over the space of different module compositions and also utilizing all three modules.
>
> > **Missing references**
>
> Thank you for these pointers! We have cited them in the updated version.

---

### Official Review · Reviewer_Bbv2 · 2022-10-27

**Confidence:** 4
**Correctness:** 4
**Technical Novelty And Significance:** 3
**Empirical Novelty And Significance:** 3
**Recommendation:** 6

**Clarity, Quality, Novelty And Reproducibility:**

This paper attempts to explore the interpretability of abstractive summarization with neural modular tree. An efficient best-first method is presented based on human-written abstractive summary to answer whether SPs could effectively present the summarization process; then a summarization program generation model is proposed to generate SPs from a source document. The proposed model could help to simulate abstractive summarization behavior, and also improve the interpretability of the summarization model.

**Strength And Weaknesses:**

It is nice to propose a novel framework to explore the interpretability of abstractive summarization on a public benchmark dataset. The proposed framework is novel. But some explanations are lacking in this paper,
1. why neural modular tree could be employed in this paper? Could you give some explanations?
2. In Appendix A, the top-k sentences are selected by computing ROUGE-1 with a summary. Why is "ROUGE-1" used as the evaluation metric?

**Summary Of The Paper:**

This paper proposes an explainable abstractive summarization framework by exploring the two research questions: RQ1 and RQ2. Experiment results on CNN/DailyMail dataset are given to demonstrate the effectiveness of the proposed SP framework.

**Summary Of The Review:**

This paper proposes a summarization program, which is an interpretable modular framework to explain the abstractive summarization. SP-SEARCH could identify SPs for human-written summaries, then a summarization program generation model is generated with SPs. Experiment results are given to demonstrate the effectiveness of the proposed approach.

---

> ### Author Response · Authors · 2022-11-18
> **Response to Reviewer Bbv2**
>
> We appreciate your time, effort, and detailed comments on our paper. We would also like to thank you for acknowledging the contributions of our work. Please find the answers to your questions below:
>
> > **Why neural modular tree could be employed in this paper? Could you give some explanations?**
>
> The premise of our paper is that humans typically summarize documents in a stepwise manner by combining multiple modular operations that involve selecting relevant information, abstracting knowledge through vocabulary generalization, fusing information from multiple sentences, etc. While most of the state-of-the-art methods for abstractive summarization take an end-to-end approach which may compromise interpretability (as elaborated in the first line of the abstract and first paragraph of the introduction), we took a different route by encoding the transparent human-like summarization process through neural modules which when combined take the shape of a tree. The term “neural modular trees” is motivated by past works on neural modular networks [1, 2] and has three parts to it – “neural”, “modular” and “trees”. Our modules are “neural” because they are fine-tuned LMs. Our modular network takes the shape of a “binary tree” because of the fusion module which is a binary operation but more generally, depending on the choice of modules, these could be any “n-ary tree” or even just a linear chain, as a special case.
>
> [1] Neural Module Networks. Jacob Andreas, Marcus Rohrbach, Trevor Darrell, Dan Klein. CVPR 2016
>
> [2] Neural Module Networks for Reasoning over Text. Nitish Gupta, Kevin Lin, Dan Roth, Sameer Singh, Matt Gardner. ICLR 2020
>
>
> > **In Appendix A, the top-k sentences are selected by computing ROUGE-1 with a summary. Why is "ROUGE-1" used as the evaluation metric?**
>
> Thanks for the interesting question! We did conduct this experiment in the past and found that our search algorithm is fairly robust to what ROUGE metric is used to select the top-k initial sentences (IS). As shown below, R-2 does slightly better than R-1 and R-L but overall, we still obtain sufficiently high oracle results of 40-41 R-2 through SP-Search, irrespective of the selection metric. The top-k sentences provide a small candidate set to build the Summarization Programs on. Hence, we chose to use a more surface-level metric like ROUGE-1 instead of stricter metrics like ROUGE-L. We have updated the paper with this new result in Appendix C and Table 8.
>
> | IS Metric |     R1    |     R2    |     RL    |   RLSum   |
> |-----------|:---------:|:---------:|:---------:|:---------:|
> | R1        | 61.88     | 40.11     | 58.46     | 60.32     |
> | R2        | **62.07** | **41.61** | **59.05** | **60.76** |
> | RL        | 61.79     | 40.36     | 58.66     | 60.38     |

---

> > ### Author Response · Authors · 2022-11-29
> > **Response to Reviewer Bbv2**
> >
> > Dear reviewer, this is a gentle reminder for our response above to your questions. We would very much appreciate it if you could let us know whether we have answered all of your questions. If there are additional questions or clarifications, please let us know and we will be happy to answer those as well. If you are satisfied with our responses, please consider updating your scores. Thanks again!

---

### Decision · Program_Chairs · 2023-01-20

**Decision:**

Accept: poster

**Justification For Why Not Higher Score:**

all high

**Justification For Why Not Lower Score:**

NA

**Metareview: Summary, Strengths And Weaknesses:**

The idea of neural modular trees (adopted from neural modular nets) for interpretable summarization is interesting and novel. It shows a new way for abstractive summarization that can influence future work. I also found the SP-SEARCH module interesting and significant. I am in favour of accepting this paper.

**Note From Pc:**

if the above contains the word "oral" or "spotlight" please see: "oral" presentation means -> notable-top-5% and "spotlight" means -> notable-top-25%. As stated in our emails, we are disassociating presentation type from AC recommendations

**Summary Of Ac-Reviewer Meeting:**

NA